# The Challenges of Mitigating Climate Change Hidden in End-User Beliefs and Expectations

Gerda Žigienė *[ID], Egidijus Rybakovas [ID], Edita Gimžauskienė and Vaidas Gaidelys

Faculty of Economics and Business, Kaunas University of Technology, Gedimino Str. 50,
LT-44239 Kaunas, Lithuania; egidijus.rybakovas@ktu.lt (E.R.); edita.gimzauskiene@ktu.lt (E.G.);
vaidas.gaidelys@ktu.lt (V.G.)
* Correspondence: gerda.zigiene@ktu.lt

**Abstract:** This research explores the potential challenges of reducing climate change hidden in the personal and collective energy use-related beliefs and expectations of end users. The study proposes a new typology of social environments, using the concept of personal and collective efficacy, which is suitable for exploring the level and nature of the challenges of solving social problems that require engaging whole societies. We use empirical data from round eight of the European Social Survey, which covers more than 20 European countries, and we employ the basic statistical methods of descriptive statistics, linear correlation and population proportion. The findings suggest that the challenges to climate-change mitigation by changing energy-use behaviour could be hidden in contradictions between beliefs in personal and collective abilities to contribute and positive outcome expectations. This opportunity could be addressed by relevant policy measures, providing more evidence of positive outcomes, even from personal contributions, and developing suitable means for collective contributions to increase awareness and belief in collective engagement.

**Keywords:** climate change; self-efficacy; end-user; energy behaviour

## 1. Introduction

"The answer to the questions: 'Why do people act environmentally and what are the barriers to pro-environmental behavior?' is extremely complex" [1] (p. 240). In the latest report on global warming, the Intergovernmental Panel on Climate Change claimed that human-induced global warming reached approximately 1 °C above the pre-industrial period levels (1850 to 1900 is used as an approximation of pre-industrial temperatures) in 2017, increasing at 0.2 °C per decade, and that global warming is likely to rise by 1.5 °C between 2030 and 2052 if it continues to increase at the current rate [2]. Governments, of course, recognise the urgent need not only to adopt policy measures for businesses and societies but also to encourage individuals to become active players in reducing global warming. Efforts to include individuals might be implemented by increasing public awareness, but the gap between communication campaigns and real changes in individual behaviour to mitigate climate change remains significant. Individual engagement might be achieved by a deliberate motion from understanding to being concerned and finally to changing behaviour, which is not secured by only providing information, even if it is significant. As stated by Lorenzoni et al., "it is not enough for people to know about climate change in order to be engaged; they also need to care about it, be motivated and able to take action" [3] (p. 446).

Individuals' values, attitudes, and emotions related to certain ends as self-motivating factors can be conceptualised as efficacy beliefs [4] (p. 134), which are "beliefs in the effectiveness of personal and others' actions [to] contribute to a particular outcome or goal". Efficacy beliefs, as generalised theoretical constructs, can be decomposed into a range of beliefs and expectations and, respectively, operationalised. This research: (1) takes beliefs in personal and collective abilities to contribute and the positive outcomes of

personal and collective contribution expectations (hereafter abbreviated as B-PC-AC/PO-PC-E) as the constructs of efficacy beliefs; (2) develops a conceptual typology of the social environments defined by levels of B-PC-AC/PO-PC-E; and (3) describes certain countries' domicile categories in terms of their social environments. Overall, four types of social environments characterise societies according to inspiration and motivation for behaviour, which is required for social goals. A lack of inspiration is considered a potential challenge to social goals. The collective goal in this study is climate-change mitigation by altered end-user energy-use behaviour.

This paper's conceptual novelty is its proposed typology of social environments defined by levels of B-PC-AC/PO-PC-E. This typology of social environments shifts the concept of efficacy belief to macro-level social analytics. The conceptual proposition of this research, which is based on the concept of efficacy beliefs, is that decisions of end users to limit their energy use to reduce climate change could be motivated and thus explained by B-PC-AC/PO-PC-E.

However, studies on B-PC-AC/PO-PC-E as drivers of human energy-use behaviour to reduce climate change are still scarce. Some climate-change mitigation barriers covered by B-PC-AC/PO-PC-E indicators have been found by qualitative studies, such as uncertainty and scepticism, social norms and expectations, and denying personal contribution to climate change [3] (p. 451), but remain unexplored quantitatively in a broader geographical scope. Thus, the present study fills this empty niche by exploring the relationships among B-PC-AC/PO-PC-E indicators, which measurably manifest various efficacy belief constructs as factors and drivers of end users' behavioural changes that influence climate change.

"End users" here are personal and household users of energy-contrasting commercial users. The focus and emphasis on end users as a principal research object is determined by the concept of efficacy beliefs, which considers individual actors and the European Social Survey (ESS), to which the only respondents are individuals.

This research explores possible challenges explained by the beliefs and expectations underlying end users' decisions to limit their energy use to mitigate climate change.

The objectives of the study are as follows:

1. To conceptualise the typology of social environments that challenge or inspire collective-goal behaviour
2. To operationalise B-PC-AC/PO-PC-E as indicators of challenges to and inspirations of climate change–mitigating behaviour
3. To define social environments according to challenges to and inspirations of climate change–mitigating behaviour in various countries and domicile category samples based on data from round eight of the European Social Survey (ESS8)

The two principal constructs of the study—B-PC-AC/PO-PC-E regarding climate change—are operationalised with selected ESS8 indicators. ESS8 was conducted from 19 September 2016 to 28 December 2016 in 23 countries. This particular survey round contained a special theme addressing climate change.

From a conceptual point of view, the research proposes a typology of so-called typical social environments defined by B-PC-AC/PO-PC-E variables. These include four typical social environments: inspired (found if both beliefs in the ability to contribute and positive outcome expectations are high), unmotivated (the opposite of inspired), uncertain, and pessimistic (found if beliefs or expectations are either low or high). These conceptual outcomes can be used to explore public attitudes towards many social problems and goals, not just climate-change mitigation.

The research question of the study is this: How much are societies in surveyed countries' various domicile categories challenged by low levels of B-PC-AC/PO-PC-E regarding climate change reduction?

The following chapter reviews the relevant literature and ends with a theoretically defined typology of social environments based on levels of B-PC-AC/PO-PC-E. Section 3 describes the data source and provides reasoning for the operationalisation of the theoretical



constructs, and it explains the data-processing and statistical methods. Section 4 presents the results and empirical evidence, and Section 5 discusses the implications of the results.

## 2. Literature Review and Theoretical Background

In recent years, public debates have started to address how best to engage public participation to reduce climate change, ranging from communication efficiency about the issue to ways of involving society and individuals. Initiatives to reduce climate change may occur primarily through government regulations or individual behaviour change, but research suggests that encouraging attitudinal change alone is unlikely to be effective [5,6]. Overall, three ways to engender mitigative behaviours by governmental regulations are enforcing green behaviour, encouraging voluntary behaviour change using economic incentives [6], and offering intensive environmental education. However, these top-down approaches alone are not enough to achieve significant results without individuals' firm beliefs in the efficacy of their actions: "mitigation policies risk being ineffective or rejected by a public lacking an understanding of the issue" [3] (p. 446).

Individual behaviour certainly plays a critical role in encouraging societal actions that mitigate climate change [7–9]. Communication and policy measures are paying increasing attention to systemic change of individual behaviour to encourage individual climate change-reducing actions [10,11]. Policymakers and environmental educators consider various forms and means of communication to strengthen the informational impact on people's environmental behaviour changes. Although the old, simple linear model—more information leading to changes in attitude and environmental behaviour—appears not to work, many governments and scientific non-governmental organisations continue merely to provide facts about climate change and expect self-contained shifts in pro-environmental individual behavioural change [1,12–14].

Indeed, individuals face not only information asymmetry but also the structural constraint of most urban infrastructure ignoring energy-sustainability principles, which hinders individual action to reduce personal carbon footprints [6]. Other incentives of pro-environmental behaviour are direct personal benefits (e.g., lower electric bills) and the palpable impact of collective behaviour change. Individuals might lack the belief that other individuals contribute as much as they do [15], however, which might discourage them from acting to address collective problems, lowering their individual belief as well [5]. Looking for desired community-wide outcomes, such as energy-demand reduction, attention should be paid to the main drivers, such as beliefs and expectations behind desired actions, decisions, and behaviours. "Beliefs in the effectiveness of personal and others' actions contribute to a particular outcome or goal" [16] (p. 16), driving human preferences. As stated by Watabe and Gilby, "Systemic changes for sustainable living are [...] not about simply improving people's awareness or attitudes [...]. They are the creation of capacities and aspirations of people actively and continuously engaging to shape alternative systems of living" [17] (p. 1). The recent shift in the scientific literature from individual pro-environmental behaviour [18–20] to collective or systemic pro-environmental behaviour [17,21–23] also indicates the importance of ties between individuals and groups in fostering climate-change mitigation.

Individual and public engagement with climate change issues is a broad, widely discussed topic. Several conceptual and empirically grounded theories compete to explain the factors behind individual and collective contribution and engagement. Engagement with climate change and a willingness to contribute with respective behaviour can largely be explained by economic factors, for example [24]. Economic recessions were confirmed to limit interest in and concern for climate change. The range of considered factors also includes "demographic factors, external factors (e.g., institutional, economic, social, and cultural) and internal factors (e.g., motivation, pro-environmental knowledge, awareness, values, attitudes, emotion, responsibilities, and priorities)" [1] (p. 239). In this context, rational choice theory would suggest that individuals' self-interest in expected outcomes is the most relevant motivating factor [25]. Reduced costs and other factors that increase

accessibility to climate change-relevant resources and facilities could also encourage more engagement [25].

Individuals' values, attitudes, and emotions related to ends as self-motivating factors are conceptualised as efficacy beliefs [26], which "provide people with a self-motivating mechanism that mobilizes effort to direct behavior toward goals and to increase persistence over time" [4] (p. 133). Efficacy belief-driven motivation is based on "beliefs in one's capabilities to organize and execute the courses of action required to produce given attainments" [4] (p. 133). Climate change and overall environmental issues on collective and macro-social levels require the consideration of engagement-motivating factors, not only on the individual but also on the collective level. Thus, beliefs not only in personal or individual efficacy but also in collective efficacy should be considered.

People's motivation to engage and contribute is defined by efficacy as "the ability, especially of a medicine or a method of achieving something, to produce the intended result" [27]. "Self-efficacy is expected to be associated with personal outcome expectancy [...], [that is], people's beliefs about their capabilities to produce designated levels of performance " [16] (p. 16). Bandura [28] initially explained the concept of self-efficacy as a personal judgement of how individuals can implement actions required to deal with future situations. Later, the self-efficacy concept was developed into the framework of collective efficacy [26], considering that, beyond individual efficacy, there is a collective belief that the group can accomplish tasks, which cultivates collective-citizenship behaviour [16,28]. Bandura [26] proposed two important attitudes for strengthening beliefs in self-efficacy: (1) social models that encourage individuals witnessing their peers succeed in comparable activities to believe in the positive results of their activities; and (2) social persuasion, which strengthens individual beliefs.

Efficacy beliefs as motivating factors and drivers in collective problem solving are twofold: beliefs in the efficacy of personal and collective actions. These two can be further separated into beliefs that individuals and collectives can contribute to solutions and expectations that personal and collective actions will contribute to expected solutions and final goals [16]. So, efficacy beliefs comprise beliefs in the ability to contribute and expectations of positive outcomes of doing so.

The efficacy concept [4,16,26,28] is associated with the ability to achieve intended results [27]. Later, this article uses the terms "belief in self and collective ability to contribute" instead of "belief in self and collective efficacy". Though the underlying conceptual reasoning is based on the efficacy concept, the term "ability" is more common and will be more informative in discussing and presenting research conclusions.

The conceptual assumption and main theoretical proposition driving this research are that challenges to changes in energy-use behaviour to reduce climate change may be hidden in individuals' various efficacy beliefs. It is proposed that such challenges might emerge when beliefs in personal and collective abilities to contribute and anticipated outcome expectations are low, resulting in low engagement and low motivation to mitigate climate change.

This construct (i.e., expectations about collectively achieved outcomes and beliefs in collective abilities to contribute to expected outcomes) and the significance of these phenomena in social interactions and social exchange are also supported by the foundations of social theory as proposed by Comelan [29], who made fundamental theoretical assumptions about "bringing participants together" for collective goals. This is consistent with social practice theory [13], which treats individuals as agents who perform many socially accepted practices. The main idea is that social interactions (which could also be social agreements to contribute collectively to climate change reduction) are possible and meaningful when they lead to the realisation of the individual interests of micro-level participants acting in macro-level social systems. Norms in the form of institutions are formed and accepted to have relevant means to forecast the actions of other participants and to attribute the expectations about those actions' outcomes. When individual inter-

ests are realised through cooperation, many individuals place stakes on such interactions, becoming so-called "stakeholders".

Following stakeholder theory, climate-change mitigation should be an interest of individual actors who understand that their interests cannot be realised using only individual resources. Only then will beliefs in collective contribution and respective outcome expectations become relevant motivators of personal contribution. Beliefs in the collective ability to contribute to climate-change mitigation are then expected to be associated with norms, community-wide agreements, and even lows that ensure collective contributions. These social laws are also defined by the value–belief–norm model [30].

To summarize this theoretical discussion, the conceptual framework suggests that relevant drivers motivate engagement and contribute to certain actions to solve collective (social or community-wide) problems when:

1. Actors (individuals in the context of climate change-mitigation research but also organisations generally) perceive themselves and the overall collective to be able to engage and contribute to some predefined purposeful actions or behaviour.
2. Actors expect that these personal and collective actions and contributions are efficient and that they will contribute to the anticipated solutions and collective goals (i.e., when actors feel they have stakes in the expected outcomes).
3. Beliefs in personal ability to contribute correlate with positive outcome expectations and when beliefs in collective ability to contribute correlate with positive outcome expectations.

These factors on both the personal and collective layers are fundamental conditions for any collective outcome-sensitive behaviour. We define four typical social environments based on the dimensions of beliefs in personal and collective ability to contribute and positive outcome expectations, that is, within the B-PC-AC/PO-PC-E framework (Figure 1). Personally and collectively inspired end-user communities are expected to act on collective goals. When the members of inspired communities living in an inspiring social environment believe that they can contribute, the overall community (i.e., the collective) can also contribute, and this contribution will deliver expected results. However, personal and collective inspirations based on beliefs in the ability to contribute and positive outcome expectations do not necessarily coincide; other, less favourable types of social environments can play roles.

Mismatches between beliefs in the ability to contribute and outcome expectations on personal and collective levels (here defined as uncertain and pessimistic end-user communities) should be considered in encouraging end-user behaviour to achieve collective goals. Such encouragement is more effective when it considers common levels of belief in the ability to participate in and achieve desired collective outcomes. Uncertain communities (Figure 1) are better inspired by increasing their beliefs in personal or collective contribution abilities. Pessimistic ones, in contrast, require visible evidence pointing to benefits and real outcomes from both personal and collective engagement. Unmotivated communities can be inspired in both ways. The implications of this conceptual framework will be discussed in greater depth at the end of the paper on the basis of the empirical results of the research.

Following the value–belief–norm model [30], individual values combined with beliefs about climate change and feelings of personal responsibility promote personal preferences for energy sources and energy-demand reduction [16]. Personal energy-demand reduction that reduces climate change is an expected collective (i.e., macro) level outcome based on end users' individual behaviour. According to the theoretical considerations discussed above, beliefs in individual and collective abilities to contribute to expected macro-level outcomes (or self-efficacy) and expectations about positive outcomes from personal and collective contributions are relevant factors driving individual behaviour and leading to collectively desired outcomes (Figure 2). The B-PC-AC/PO-PC-E framework and the proposed typology of social environments are located in a broader picture of cause–effect relationships and the potential of explanatory research and energy use–behaviour modelling. This study covers B-PC-AC/PO-PC-E framework–based social environments and

construct operationalisation but does not explain the actual behaviour of energy end users in certain societies.

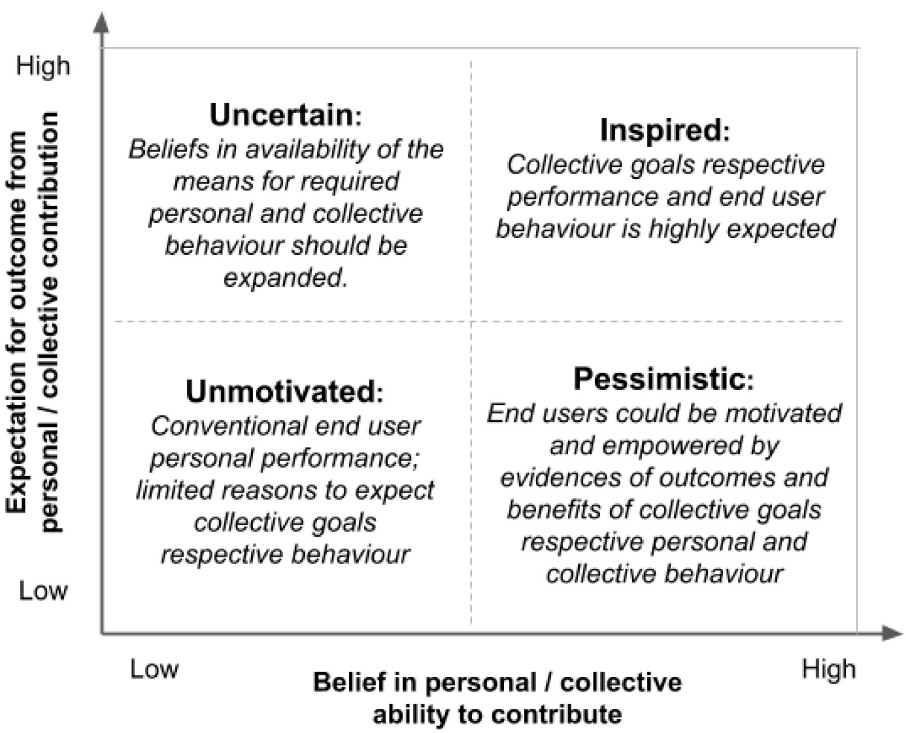

**Figure 1.** Typical social environments inspiring and motivating collective goal-respective behaviour.

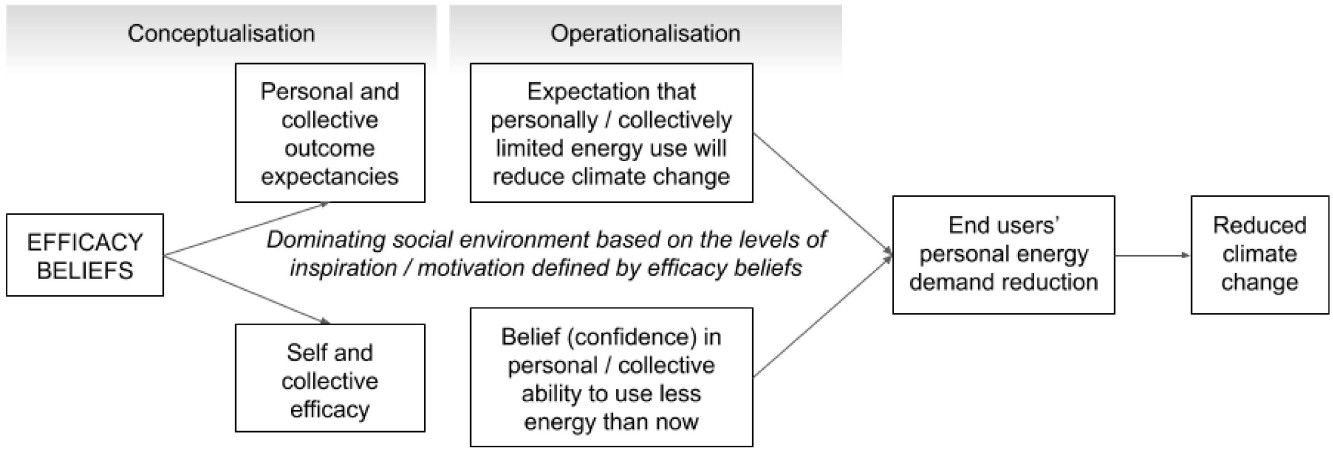

**Figure 2.** Beliefs in the ability to contribute and positive outcome expectations as drivers of end-user behaviour in the context of climate-change mitigation.

The challenges to reducing climate change, which this paper treats as hidden, can be revealed empirically by observing low levels of the B-PC-AC/PO-PC-E framework's measuring indicators. A negative, weak, or non-existent relationship between any two B-PC-AC/PO-PC-E components is more evidence of challenges in the social environment of a certain community. In contrast, high levels of B-PC-AC/PO-PC-E components, along with a strong association between beliefs and expectations, indicate a positive context for climate change mitigation-oriented energy-use behaviour with fewer challenges. Beliefs and expectations are not manifested in everyday living because they are not observable behaviour,

which is why low levels of beliefs and expectations are treated as hidden challenges. These challenges are explored and described empirically in the following chapters.

## 3. Materials and Methods

This empirical research is descriptive, not explanatory. It describes selected countries and their domicile categories by their dominant types of social environments (Figure 1) and their levels of inspiration and motivation based on B-PC-AC/PO-PC-E following the theoretical considerations and conceptual framework presented above. The research does not try to explain the actual performance of certain community members caused by their beliefs and expectations but only to describe the features of communities in selected countries covered by ESS8. B-PC-AC/PO-PC-E components are operationalised with four ESS8 module variables from "Public Attitudes to Climate Change, Energy Security, and Energy Preferences" [16,31] (Figure 3).

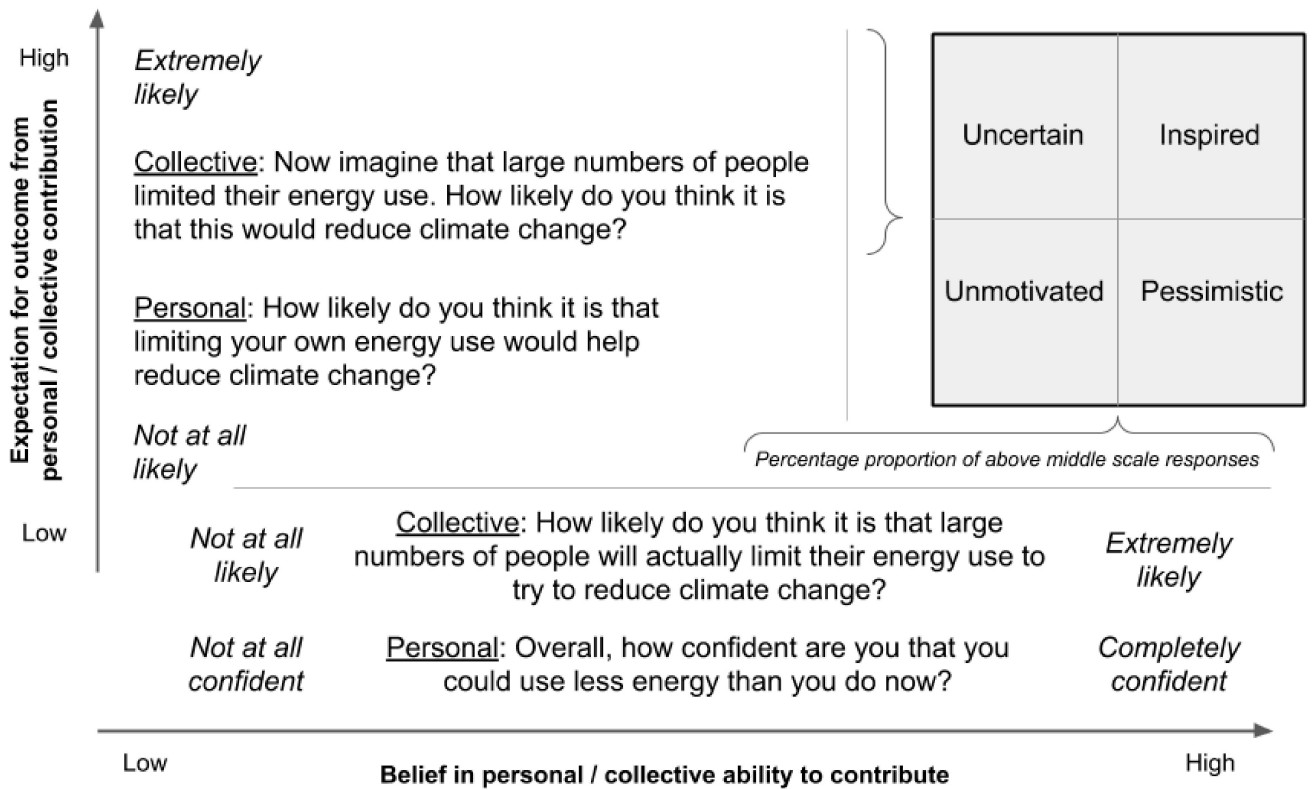

**Figure 3.** Operationalisation of theoretical framework with the European Social Survey variables.

The social environment of a country and a certain living area (domicile) category is considered inspired to reduce end-user energy demand if it is dominated by individuals who believe both in their personal ability to reduce energy demand and in the abilities of others and who expect positive effects from climate-change mitigation both from their personal and from collective engagements (Figure 3). The lack of these beliefs and expectations gives rise to environments defined as uncertain, pessimistic, or unmotivated to reduce energy demand and thus mitigate climate change.

These theoretical considerations and respective propositions operationalised with the ESS8 variables generate the following objectives of this empirical research:

1. To measure beliefs in personal and collective abilities to contribute to climate-change mitigation and positive outcome expectations in the surveyed countries and various domicile categories
2. To evaluate the proportions in surveyed societies of people with certain beliefs and expectations

3. To determine the dominant type of social environment according to considered beliefs and expectations in the surveyed countries and various domicile categories

To evaluate the correlations between beliefs in personal and collective abilities to limit energy use and positive-outcome expectations, ESS8 data [31] in the ESS database were accessed on its webpage. All survey round data are publicly available based on individual registration. The ESS covers more than 20 European countries and Israel. The ESS8 covers Austria (AT), Belgium (BE), Switzerland (CH), Czechia (CZ), Germany (DE), Estonia (EE), Spain (ES), Finland (FI), France (FR), United Kingdom (GB), Hungary (HU), Ireland (IE), Israel (IL), Iceland (IS), Italy (IT), Lithuania (LT), the Netherlands (NL), Norway (NO), Poland (PL), Portugal (PT), the Russian Federation (RU), Sweden (SE), and Slovenia (SI). It covers countries at different stages of socio-economic development, with political regimes and traditions, with cultural and historical backgrounds, and in different geographies. Though formal statistical hypotheses are not stated for descriptive analyses, such a wide country coverage should help us observe country group–related patterns of relations within the B-PC-AC/PO-PC-E structure.

The domicile variable includes five types of residential areas. Respondents were asked to select their type of residential area based on their own judgement. The options offered were a big city (C), the suburbs or outskirts of a big city (S), a town or small city (T), a country village (V), a farm (F), and a home in the countryside [31]. The empirical data analysis was done by separate country and domicile category subsamples.

Weighted data were used to calculate all the statistical estimates. The ESS provides design, post-stratification, and population size weighting variables [32]. The post-stratification weight included a design weight that corrected data for potential sampling design and non-response biases, which was the most appropriate weighting method considering the design of the present research.

This three-dimensional analysis allows us to triangulate the research, increasing the reliability of the results and conclusions. The structure of the research design is as follows:

1. Averaged levels of B-PC-AC/PO-PC-E indicators in samples from different countries and domicile categories were measured; sample mean values were used to describe sample data, and 95% confidence intervals (CI) for mean values were calculated for population value estimates and population comparisons.
2. Variables were dichotomised to observe the proportions of individuals in each sample and respective population who value their beliefs and expectations relatively highly. Original [31] Likert-scale scores responses from 6 to 10 (on a total 0 to 10 scale) were considered to indicate relatively high evaluations of beliefs and expectations. CIs for proportions were calculated for population proportion estimates.
3. Calculations of correlations between beliefs in personal ability to contribute and respective outcome expectations and beliefs in collective ability to contribute and respective outcome expectations were also made. A stronger association between beliefs in the ability to contribute by limiting energy use and climate-change mitigation outcome expectations indicated a better balance between beliefs and the community's degree of inspiration.

Samples containing fewer observations than a predefined number of valid cases were not considered. The rule of thumb in statistics is that samples of 30 observations are sufficient for statistical analysis applying conventional theories, such as the Central Limit Theorem and inference methods. Since it is not expected that data are normally distributed, no other formal requirements for the distributions were set.

The ESS8 variables "cflsenr", "ownrdcc", "lklmten" and "lkredcc" (which are operationalised to measure constructs in B-PC-AC/PO-PC-E framework) [31] are primary data sources following the operationalisation presented above (Figure 3).

The data distribution charts in Appendix A indicate that some sample data are not normally distributed. Since the mean or other parametric characteristics are biased in abnormally distributed samples and are thus not reliable enough to describe the data,

the second step was taken of transcoding variable values into a binomial scale, which aggregates low values (scores from 0 (not at all confident, not at all likely) to 5) and high values (scores from 6 to 10 (completely confident, extremely likely)). The proportion of high-end aggregate evaluations was taken to measure the respective probability proportion in the sample and an inferential estimation of the populations of various countries' domicile categories.

The descriptive measures of the concepts of interest are given as probable proportions in the population, where respective concepts are evaluated as relatively high. The inferences to real population proportion values were made considering CIs for proportions [33,34]. The CIs are different for each subsample due to extensive variation in sample sizes (see Appendix B for sample sizes and main descriptive statistics). The Pearson–Clopper CI for proportion probability in a binomial distribution was calculated. This particular type of CI has its own pros and cons, which are discussed in the literature [35,36]. Some CIs for binomial proportions are considered to perform better with small sample data than others with larger ones, the Pearson–Clopper alternative was chosen because it uses the tail method and is common. However, note that "various evaluations indicate that the Clopper–Pearson interval tends to be extremely conservative for small to moderate n (sample sizes)" [35] (p. 297), which is the case in some of the compared countries' domicile category samples.

Dichotomisation, that is, the transformation of Likert-scale data to binomial data, was performed to determine probable population proportions; all the considered concepts were evaluated with the five highest scores (from 6 to 10). Probable population proportion values are alternative descriptive measures of the variables of interest besides the sample mean. They directly expose how wide and extensive positive attitudes towards the concepts of interest are in certain societies. The mean is sensitive to biases caused by characteristics of data distribution. The original data scale is split "at a fixed point on the scale designated a priori", which is one dichotomisation alternative [37]. The decision to split the scale into six scores was made considering the variation of dichotomous outcome variables.

Transcoding to a binomial scale was done because the cost of dichotomisation is discarding data that are necessary to compare individuals [37]. The aim of the present research, however, is to reveal community-wide attitudes measured in terms of beliefs and expectations; comparing observations on the individual level is less relevant. Comparisons between groups were also not done. However, sample means, graphical data distributions, and correlation analysis represent the full data variation. The analysis of probable proportion values is descriptive; it does not estimate impacts or relationships between concepts. Potential relationships and interdependencies are discussed based on descriptive statistics and their graphical representations.

The clear existence of two distinct types (groups or taxons) of individuals in the population should be clear to support dichotomisation decisions [37]. The clear split point also should be evident [37]. These two requirements constitute the methodological limitations of dichotomisation. If the existence of two distinct groups is not proven, it might be that between those who believe and those who do not, moderate "believers" exist in a society, as the Likert scale variables suggest. The split point is also unclear and might differ from sample to sample. These limitations should be considered in drawing conclusions on the basis of probable proportion analysis, so it is not the only method applied; graphical data representation, sample means (with a standard mean error-based CI of 95%) and correlation analysis are also done.

The correlation analysis was done for the original Likert scale variables, retaining all the collected data (not aggregated, averaged, or dichotomised). The correlations were estimated using the Pearson correlation coefficient. As already noted above, not all the samples were close to a normal distribution, so Pearson's correlation coefficient could also be biased and not always be relevant, another methodological limitation to be considered. However, the main conclusions are drawn based on mean and proportion values, so the possible correlation biases do not threaten the validity of the study's main findings.

All the statistical calculations were made with IBM SPSS version 25.

## 4. Results

A total of 23 countries and five domicile categories created a group of 115 subsamples, which were taken as the objects of empirical research. These samples represent five domicile categories in all the countries covered by ESS8. Sample sizes range from only a few to 1400 responses (Appendix B). Lithuanian suburbs and Czechia, Hungary, Israel, Iceland, Lithuania, Poland, Portugal, and the Russian Federation farm subsamples (nine in total) were excluded from further analysis due to small sample sizes. Appendix B presents the numbers of valid and missing observations, means, standard errors of mean, and standard deviations for all the subsamples and each of the four variables of interest. These basic descriptive statistics should be considered along with the graphical frequency distributions (Appendix A) since some distributions deviate from normal, possibly making parametric descriptive statistical estimates biased and not reliable enough to describe the sample data.

Figure 4 depicts sample means of B-PC-AC/PO-PC-E variables in different countries and domicile category subsamples. Measurements from both personal and collective perspectives are indicated by separate series highlighted with different colours. Most country–domicile subsamples classify as so-called pessimistic social environments if measured from the perspective of personal beliefs and respective expectations. However, the positioning of the same group deviates to an uncertain social environment when beliefs in collective contribution ability and respective outcome expectations are considered.

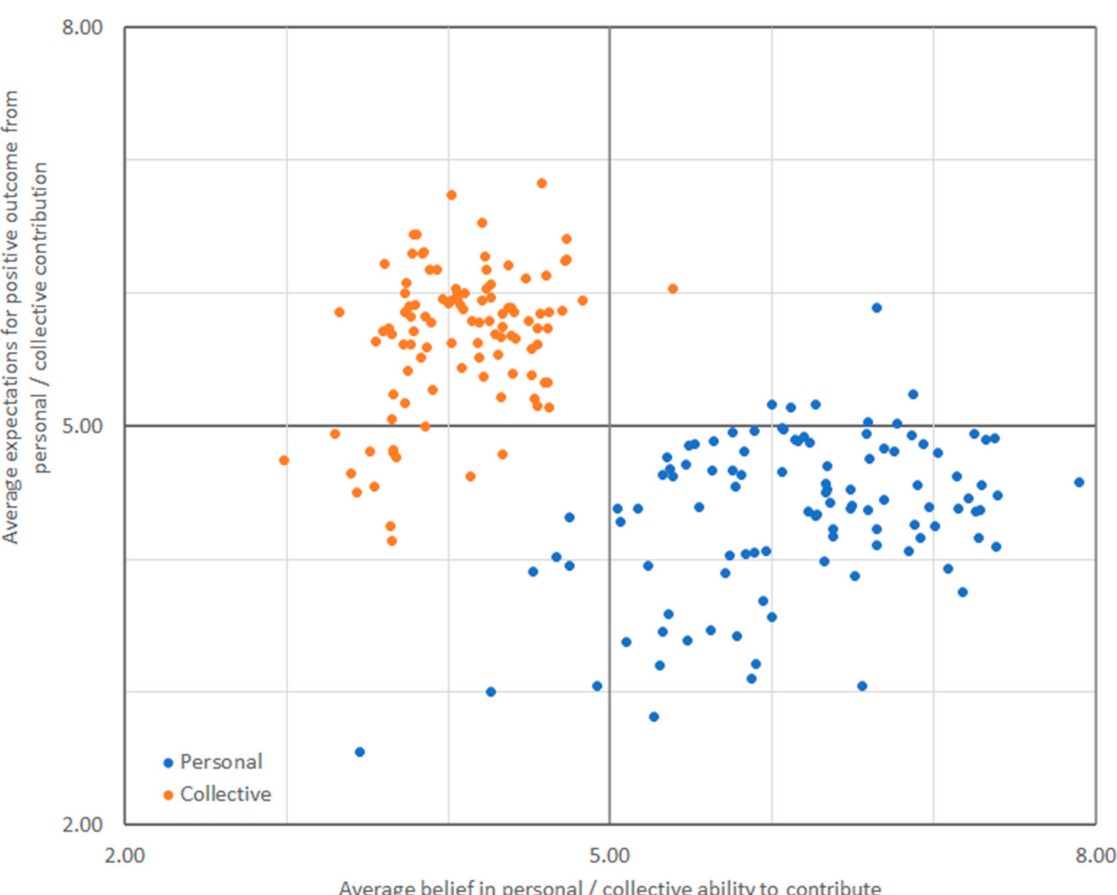

**Figure 4.** Sample means of belief in the ability to contribute to climate-change mitigation and positive outcome expectations in different countries and domicile category samples (European Social Survey round 8 data and authors' calculations).

Clearly, end users of energy resources generally believe (or are confident) that they can use less energy than they do now but do not expect that limiting their own energy use would

help to reduce climate change. However, noticeably higher expectations of the outcomes of reducing climate change are associated with outcomes of collective engagement, but belief in the ability of many people to limit their energy use to reduce climate change is much lower (Figure 4).

These conclusions should not be taken as absolute; they are merely based on statistical estimates (i.e., sample mean values). Based on these survey data, however, reduced climate change as a collective goal is generally challenged by lacklustre end-user expectations about reduced climate change as an outcome of limiting their own energy use. Another challenge is a lack of belief in collective contributions, which could also limit personal energy use.

The figures in Appendix C indicate the positioning of exact country–domicile subsamples. For example, the alignment of Figures 4 and A5a reveals that Ireland's big-city sample is the only one categorised as inspired by both perspectives: beliefs in personal and collective ability to limit energy use and expectations to successfully mitigate climate change. Hungary and Russian Federation suburbs and big city-outskirt samples (Figure A6a) are two cases categorised as unmotivated to mitigate climate change. Parts (a) of the figures in Appendix C also show the 95% CI of each variable mean, establishing high confidence in the estimation based on the sample mean and the statistical significance of the sample means' differences.

Some patterns in country positioning are observable in parts a) of the figures in Appendix C. For example, four Eastern European countries (Estonia, Chekia, the Russian Federation, and Slovenia) deviate noticeably from other countries in a country village subsample, showing the lowest sample mean values of personal outcome expectations (Figure A8 in Appendix C). Pessimistic personal outcome expectations are thus stronger (or positive outcome expectations are weaker) in these countries than in all the others. Among the suburbs and outskirts of big-city samples, these four countries are accompanied by two other representatives of Eastern Europe: Hungary and Poland. However, the CIs are much lower in the suburb samples, and the deviation is not very noticeable and so cannot be considered statistically significant (Figure A6 in Appendix C).

In general, Eastern European countries tend to be lower than Western European ones if compared by averaged beliefs in personal and collective ability to contribute to climate-change mitigation by limiting personal energy use and respective positive outcome expectations (Appendix C). This pattern does not change noticeably across different domicile categories. Country-by-country case-study analyses could certainly confirm statistically significant mean value differences and thus discuss respective country characteristics. The sample mean values in the most divergent cases might differ by two or even three points, such as Czechia vs. Sweden big city cases (Figure A5a) and the Russian Federation vs. Sweden town and small-city cases (Figure A7a). However, the primary focus of this research is to indicate more general challenges found in most countries despite varying intensity.

The fact that only one sample of more than 100 from five domicile categories in 23 countries is considered representative of communities inspired by both personal and collective end-user behaviour to mitigate climate change is sufficient proof to support the conclusion that theoretically proposed and anticipated challenges do exist and are hidden in end-user beliefs and expectations associated with contribution ability and positive climate change outcomes.

Sample mean values are sensitive to the shapes of data distribution (Appendix A) and thus might hide real characteristics that are important for conclusions and implications. Due to the many samples and differences in sample sizes, they are often insufficient for parametric statistical descriptive analysis. Data distribution frequencies and sample data based on expected population proportions are other ways to represent and compare the characteristics of different samples. Figure 5 shows the sample proportions of respondents who value their beliefs and expectations relatively highly. Parts b) of the figures in Appendix C also indicate the positioning of exact samples from different countries and domicile categories. The 95% Clopper–Pearson CIs for the proportions given in parts b) of

the figures in Appendix C let us draw conclusions about probable proportion differences and statistical significance in different populations.

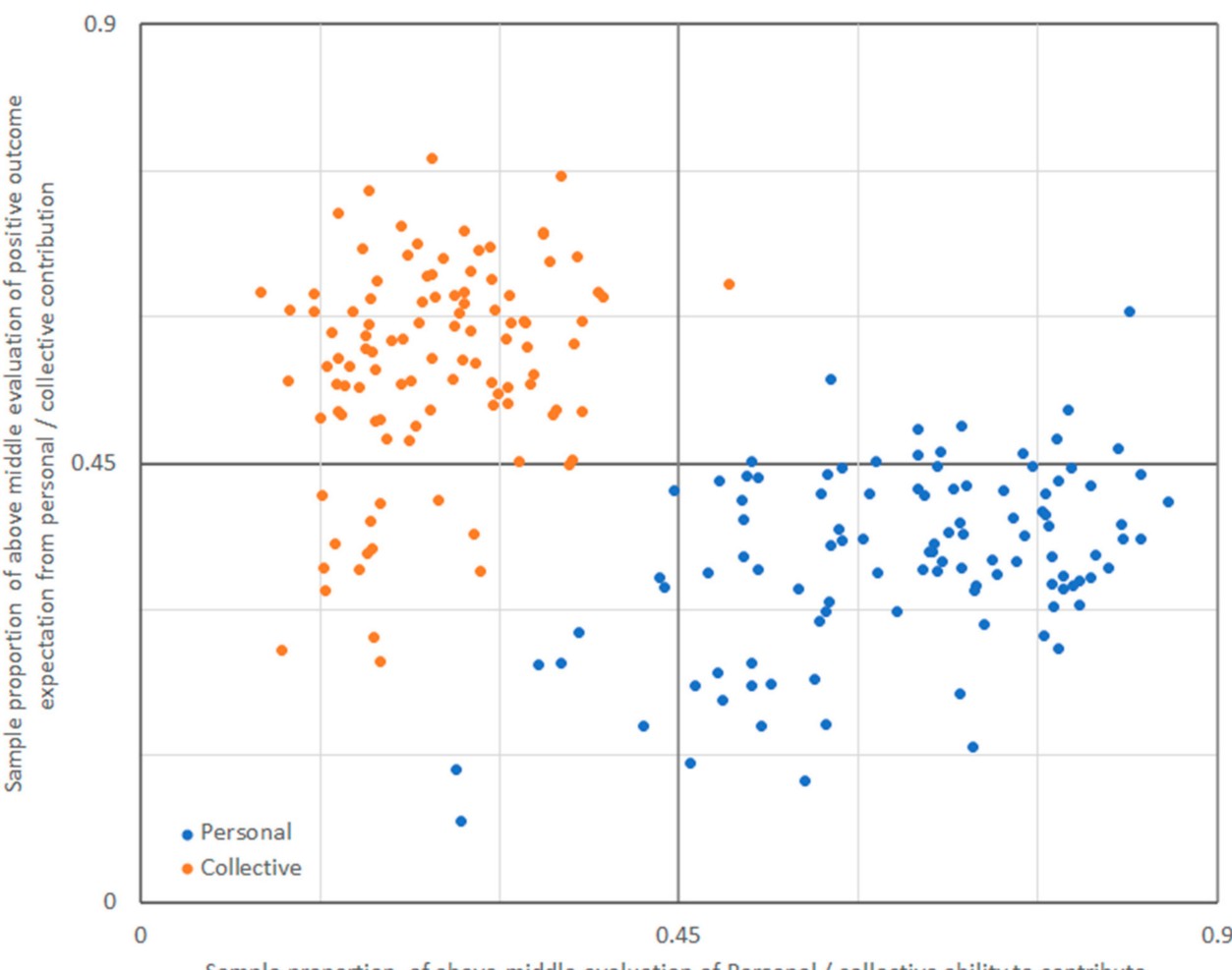

**Figure 5.** The sample proportion of answers ranging from 6 to 10 evaluating beliefs in abilities to contribute to climate-change mitigation and positive outcome expectation in different countries and domicile category samples (European Social Survey round 8 data and authors' calculations).

The sample proportions of respondents who evaluated B-PC-AC/PO-PC-E variables relatively highly, taken as descriptive indicators, do not noticeably change the overall research results in the surveyed samples. General conclusions based on proportion values are the same as those already concluded based on the sample mean values. Again, the absolute majority of the surveyed samples tend to be pessimistic about the climate-change outcomes of personal engagement and uncertain about collective contributions or collective abilities to mitigate climate change by limiting energy use (Figure 5).

The calculated sample proportion values indicate that in most of the surveyed samples, the shares of respondents who valued their beliefs relatively highly (i.e., 6–10 Likert scores) in their personal abilities to reduce energy consumption fall into the 0.30 to 0.90 interval, but the positive outcome expectations of these personal contributions are estimated relatively highly by only a 0.15–0.45 share of the respondents. Collective perspective evaluations indicate the opposite interaction between beliefs in abilities to contribute and positive outcome expectation (Figure 5): a 0.15–0.45 share of the respondents believed in a collective ability to engage, but positive outcomes from collective energy-use limitations were expected by a 0.30–0.75 share of the respondents in different country and domicile category samples.

This interaction is the same as that already discovered based on the sample mean values (Figure 4). The sample proportion indicator does not generate new insights, but it lets us estimate the expected number of individuals who value their beliefs relatively highly in the sampled populations (Appendix C). Populations' respective estimates should be made according to CIs for calculated sample proportions.

Correlations between B-PC-AC/PO-PC-E variables add insights to these conclusions (Figure 6). In total, 70 of the 106 tested samples have statistically significant Pearson correlations between beliefs in personal and collective abilities to contribute to climate-change mitigation and respective outcomes from personal and collective contribution expectation variables. The remaining 36 samples are dominated by those with statistically significant correlations only between beliefs and expectations in collective perspective variables, three samples with significant correlations only between beliefs and expectations in personal perspective variables and seven with no correlations at all. Appendix D presents the Pearson correlation coefficients, their statistical significance and the number of pairwise cases. Notably, the reliability of Pearson correlations depends on the data distribution and the extent of their deviations from a normal distribution (Appendix A).

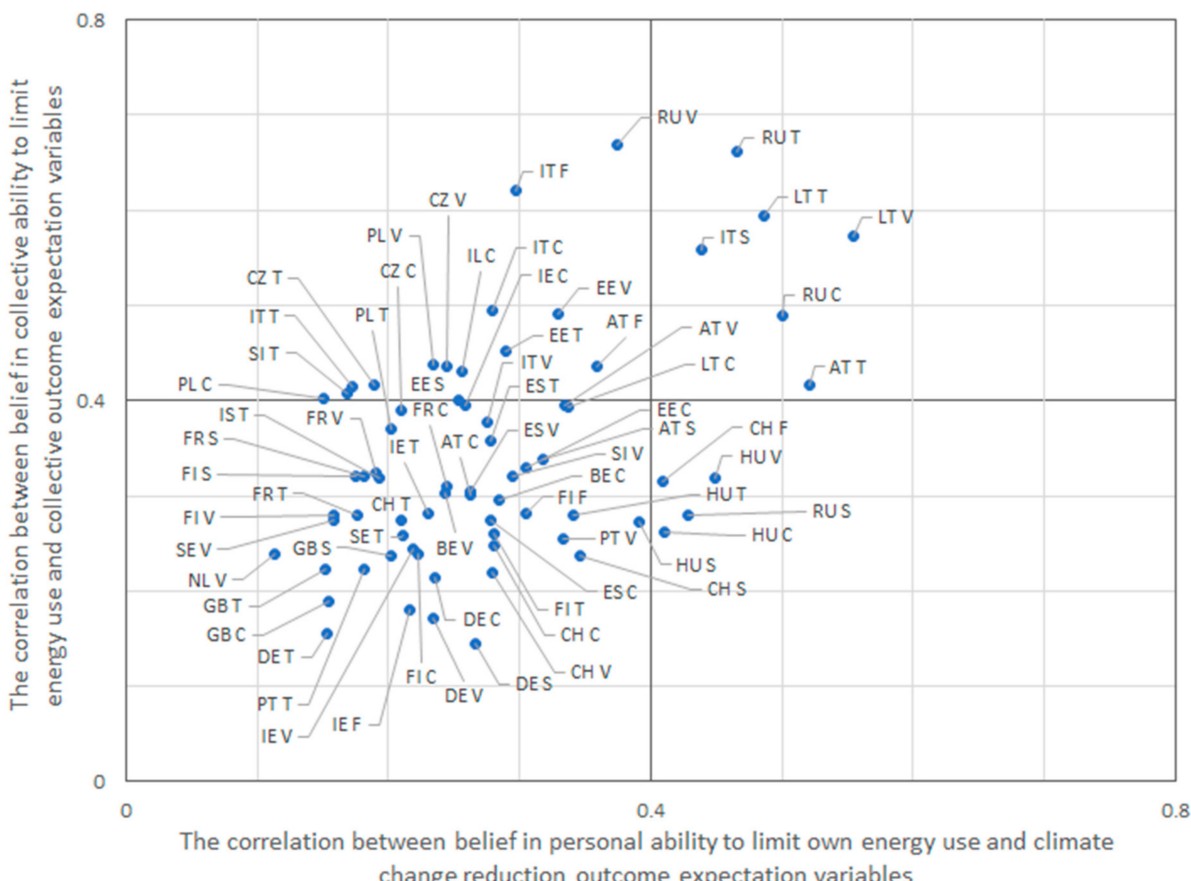

**Figure 6.** The positions of surveyed samples according to correlations between beliefs in personal and collective abilities to contribute to climate-change mitigation and respective positive outcome expectations.

Figure 6 indicates that only samples with statistically significant correlations in both personal and collective beliefs and expectation perspectives. Most of the samples are between 0.1 and 0.4 Pearson correlation coefficient values in beliefs and expectations from both personal and collective perspectives. No noticeable clustering of samples based on the domicile categories is observable.

A weak-to-moderate strength of correlation between beliefs in the ability to contribute by limiting energy use and climate-change mitigation outcome expectations is one sup-

posed outcome of the correlation analyses, which uncover once again, but in another way, the hidden challenges of mitigating climate change by changing end-user energy use–related behaviour. Lower values of the correlation coefficient indicate that changing beliefs in personal and collective abilities to contribute to climate-change mitigation by limiting personal energy use is weakly associated with the variation in expectations of positive outcomes of climate-change reduction, meaning that a lower correlation indicates a higher deviation towards pessimism (when beliefs in personal contribution and outcome expectancy are measured and associated) or uncertainty (when beliefs in collective ability to contribute and outcomes expectancy are measured and associated).

Higher values of the correlation coefficient indicate a closer association between beliefs in the ability to contribute and respective outcome expectations but do not guarantee that beliefs and expectations by themselves are evaluated highly. For example, Lithuanian towns, small cities, and country villages are high-correlation cases (Figure 6), but their evaluations of beliefs and outcome expectations themselves are only moderate (Figures A7 and A8 in Appendix C). A correlation analysis better reveals the cases where deviation towards pessimism about outcomes (individual perspective) and uncertainty about people's ability to contribute (collective perspective) is less noticeable and less dominant when correlation coefficients are higher.

In summary, the three triangulated analytical approaches (sample mean, sample proportion, and correlation analyses) confirm the conclusion that the challenges to climate-change mitigation generally hide in the beliefs and expectations measured by B-PC-AC/PO-PC-E framework variables. Following the theoretical arguments, individuals can be demotivated to contribute to climate-change mitigation by limiting their own energy use when they do not believe in the overall communal (i.e., collective) ability to limit energy use. However, the same demotivating effect can be expected due to low expectations for positive outcomes from personal contributions by limiting energy use.

The two encouraging findings are that individuals are confident in their personal abilities to limit energy demand and its usage and have high expectations that collectively limited energy use will mitigate climate change. These outcomes can be considered and employed as motivating factors. The country and domicile domains do not reveal any significant differences.

## 5. Discussion

The B-PC-AC/PO-PC-E variables measuring efficacy beliefs were found to be appropriate to explore potential challenges to mitigating climate change. The typology of social environments defined by climate change reduction-related beliefs and expectations is a proposed conceptual solution for macro-level social analysis, as it extends the concept of efficacy belief to the macro social level, where it was found suitable to explore the levels and nature of challenges in solving social problems that require engaging whole societies.

The conceptual background of the research is that low levels of B-PC-AC/PO-PC-E are challenges in engaging communities to reduce energy use to mitigate climate change. This was operationalised with variables measuring individuals' beliefs in personal and collective abilities to contribute to climate-change mitigation by limiting energy use and expectations about positive outcomes in climate-change mitigation from them.

Though this research was not intended to explain actual behaviour by hidden factors, it still revealed challenges to limiting energy use to mitigate climate change. People's motivation to limit energy use and contribute to climate-change mitigation can be limited by a lack of belief in the collective ability to contribute and low expectations about positive outcomes from personal contributions; people in the surveyed countries did not believe that collective actions could limit energy use and did not expect that their personal contribution would help mitigate climate change.

These results seem reasonable and could be explained by common sense: personally, it is not difficult to reduce energy use, but it is hardly possible that such behaviour would have any impact on climate-change mitigation, and while collective energy-limiting behaviour

would effectively mitigate climate change, it hardly seems possible. In other words, most of the surveyed energy end users were pessimistic about climate-change mitigation based on limiting their own energy use and uncertain about the contribution of large numbers of other people to reduce energy use.

Building on the general foundations of social theory [29], the empirical outcomes of the study could be interpreted as exposing that communities in the surveyed countries often lack appropriate social institutions that would govern the behaviour of energy end users. The low levels of belief in collective abilities to contribute to climate-change mitigation by limiting energy use indicates that individuals do not expect that many people would change their behaviour, which could mean that social norms or widely accepted institutions are not present. Socially accepted norms or institutions could be implemented by many means, including formal and informal incentive structures.

The challenges revealed by this study could be addressed by various environmental and climate change-mitigation measures, including education and communication policies. The most straightforward approach would be establishing social norms and widely accepted institutions encouraging limited energy use and individual behaviour to mitigate climate change. These institutions would likely increase beliefs in collective abilities to contribute. Such initiatives have already started and are being developed in various places. The results of this research support these projects with more empirical evidence.

Further steps are expected to be made by researchers in this field. Quantitative cause–effect research is needed to study how individuals' actual beliefs and expectations (i.e., energy end users) influence their actual energy-use behaviour. The primary data used for this research do not generate any explanations of any dependencies because no dependent variable was measured and the variation of independent variables was limited. Not only the survey but also the objective technical data on energy usage could be employed towards these ends. It is complicated to evaluate energy-use trends by the survey data, but external objective data may be much more informative, such as the usage of energy-efficient electric appliances, water consumption, and so on.

## 6. Conclusions

The outcomes of the research are descriptive. On the one hand, this paper found that people's beliefs in their personal abilities to contribute to climate-change mitigation by limiting their energy use is high, but their positive outcome expectancy for their personal contributions is low. On the other hand, their belief in collective ability to contribute is low, but their positive outcome expectancy for collective contribution is high. These findings remain stable irrespective of different domicile categories, suggesting that challenges to climate-change mitigation from the side of energy use could be hidden in contradictions between beliefs in personal and collective abilities to contribute and positive outcome expectations.

The direct answer to the research question is that the surveyed countries' various domiciles are challenged by low levels of belief in collective ability to contribute to climate-change mitigation and low levels of positive outcome expectations from personal engagement. These are the study's findings:

1. The end users of energy tend to believe (or be confident) that they can use less energy than they consume now, but they do not believe (or expect) that limiting their own energy use would help to reduce climate change. Noticeably higher expectations of reducing climate change are associated with the outcomes of collective engagement, but belief in the ability of many people to limit their energy use to reduce climate change is much lower.

2. Reduced climate change as a collective goal is challenged by a lack of end-user belief in reduced climate change as an outcome of limiting personal energy use. The other challenge is a lack of belief in the ability of collective contribution, which could also discourage the personal limitation of energy use.

3. According to the theoretical propositions, low belief in the efficacy of personal contributions and the ability of others to contribute could prevent end users from attempting to change their energy usage to mitigate climate change. This inconsistency between efficacy beliefs is evidence that potential challenges to climate-change mitigation exist and are hidden in end-user beliefs and expectations.

4. Eastern European countries tend to be lower than Western European ones in average belief in personal and collective abilities to contribute to climate-change mitigation by limiting energy use and respective positive outcome expectations, but these differences are observed on the basis of sample data and are not statistically significant.

5. The patterns of a country's positioning do not change noticeably across different domicile categories, meaning that observed, statistically proven average levels of belief and expectations regarding personal and collective abilities to contribute to climate-change mitigation and respective positive outcome expectations are society-wide, not differing significantly across domicile categories.

6. The study has some methodological limitations. First, the data are not normally distributed in some subsamples, so the sample averages may not be perfectly reliable estimates. Second, the Pearson correlations could be affected by the data distribution patterns. Finally, data dichotomisation to represent the structure of surveyed societies broken down by the proportions of people with positive beliefs and expectations related to climate-change reduction results in some data loss. However, dichotomised variables were found most appropriate to compare surveyed subsamples, and this limitation did not impact or significantly change the overall results of the study.

These challenges are opportunities for relevant policy measures, especially with more evidence of positive climate-change mitigation outcomes, even from personal contributions, by changing energy-use behaviour. These measures could include educational campaigns and the promotion of inclusive, sustainable consumption. Developing suitable engagement, including product and service innovations (e.g., smart platforms, incentives for replacing energy-inefficient appliances, and business model innovations) for collective contributions could increase awareness and belief in collective engagement.

A suitable approach to end-user beliefs and expectations could play a significant role in engaging the public in reducing climate change. The study outcomes, based on quantitative empirical research, provide evidence that should enable the development of policy measures to promote and encourage climate change-mitigating end-user behaviour.

**Author Contributions:** Conceptualisation: E.R., G.Ž., E.G., and V.G.; methodology: E.R. and G.Ž.; formal analysis: E.R., G.Ž., E.G., and V.G.; data curation: E.R.; writing—original draft preparation: E.R., G.Ž., E.G., and V.G.; writing—review and editing: E.R. and G.Ž.; visualisation: E.R.; supervision: E.G. All authors have read and agreed to the published version of the manuscript.

**Funding:** This research received no external funding.

**Institutional Review Board Statement:** Not applicable.

**Informed Consent Statement:** Not applicable.

**Data Availability Statement:** The data that support the findings of this study are available from European Social Survey database, https://www.europeansocialsurvey.org/data/round-index.html.

**Conflicts of Interest:** The authors declare no conflict of interest.

## Appendix A. Data Distributions

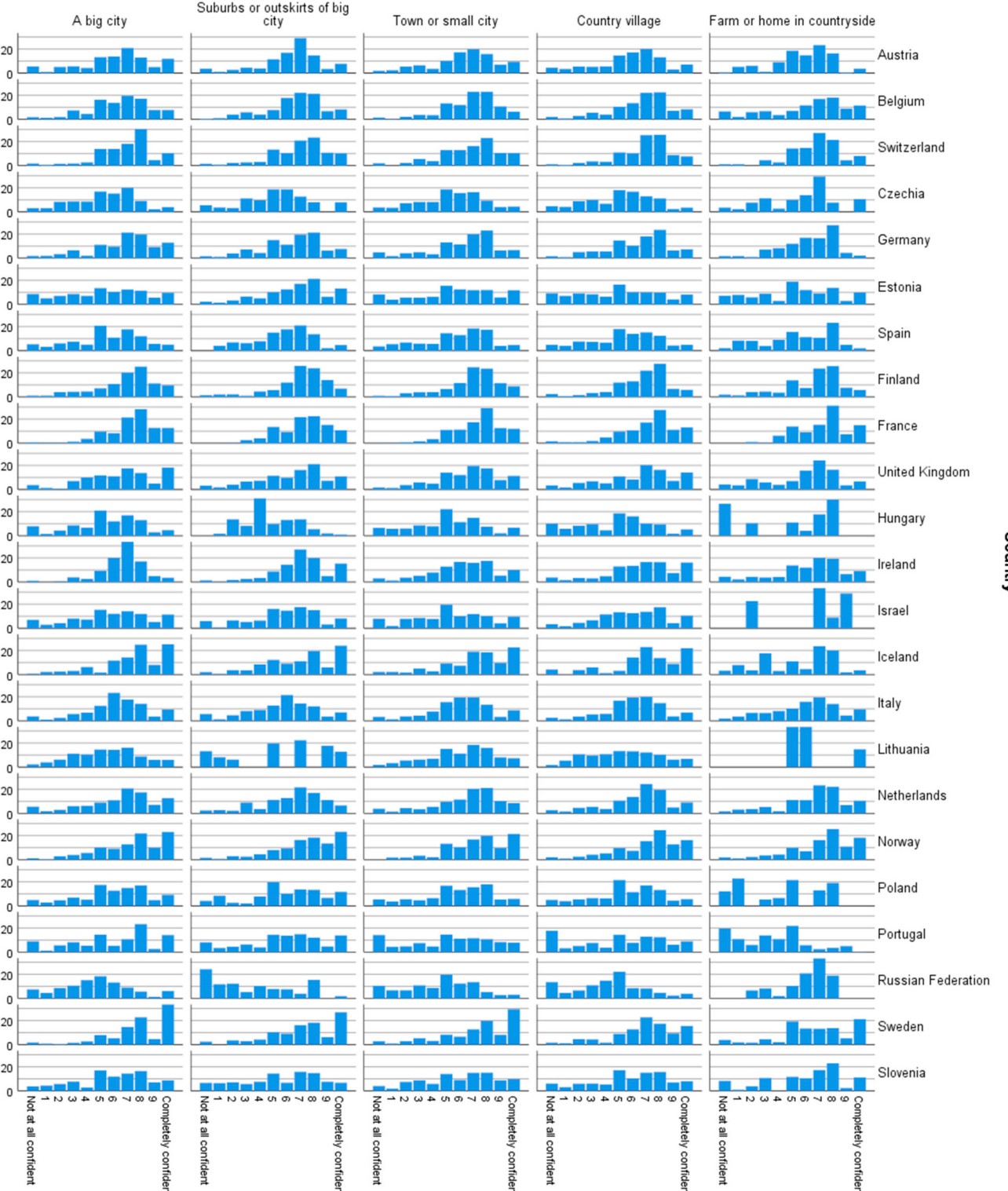

**Figure A1.** Belief in personal ability to contribute to climate change by limiting own energy use (EES8 survey question: Overall, how confident are you that you could use less energy than you do now?); percentage data distributions broken down by country and domicile categories.

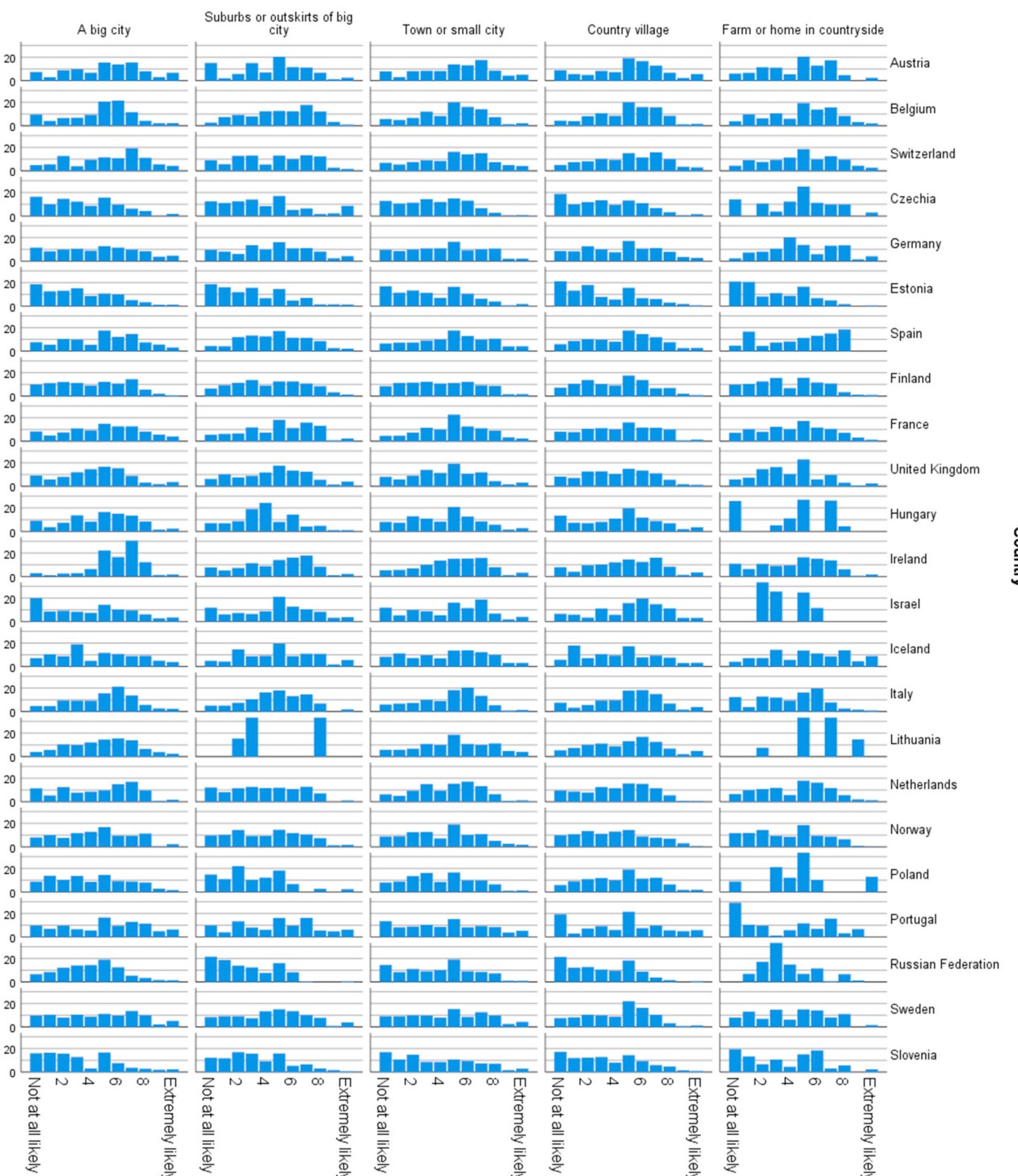

**Figure A2.** Expectation for positive outcomes from personal contribution by limiting own energy use (EES8 survey question: How likely do you think it is that limiting your own energy use would help reduce climate change?); percentage data distributions broken down by country and domicile categories.

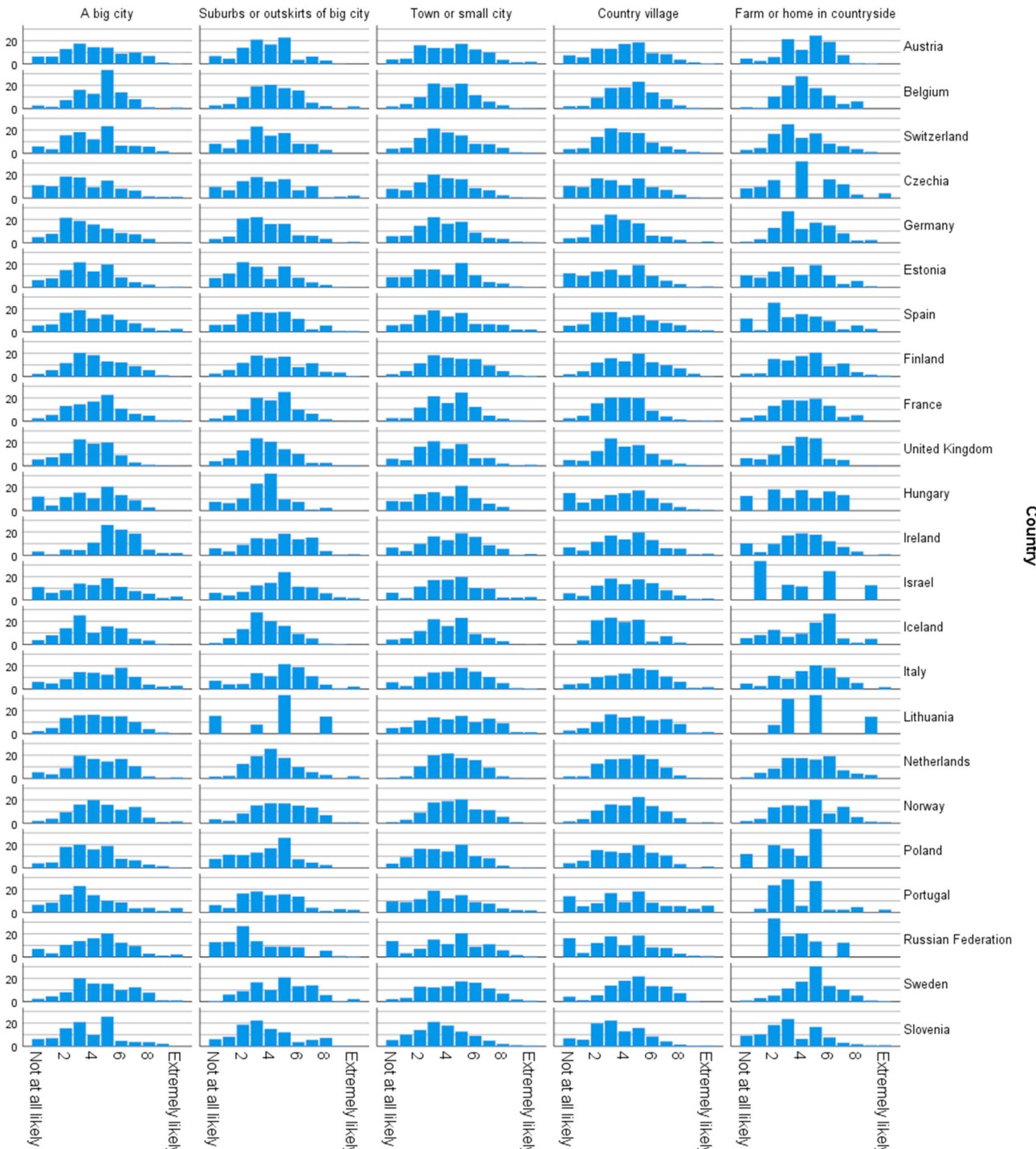

**Figure A3.** Belief in collective ability to contribute climate change mitigation by limiting energy use (EES8 survey question: How likely do you think it is that large numbers of people will actually limit their energy use to try to reduce climate change?); percentage data distributions broken down by country and domicile categories.

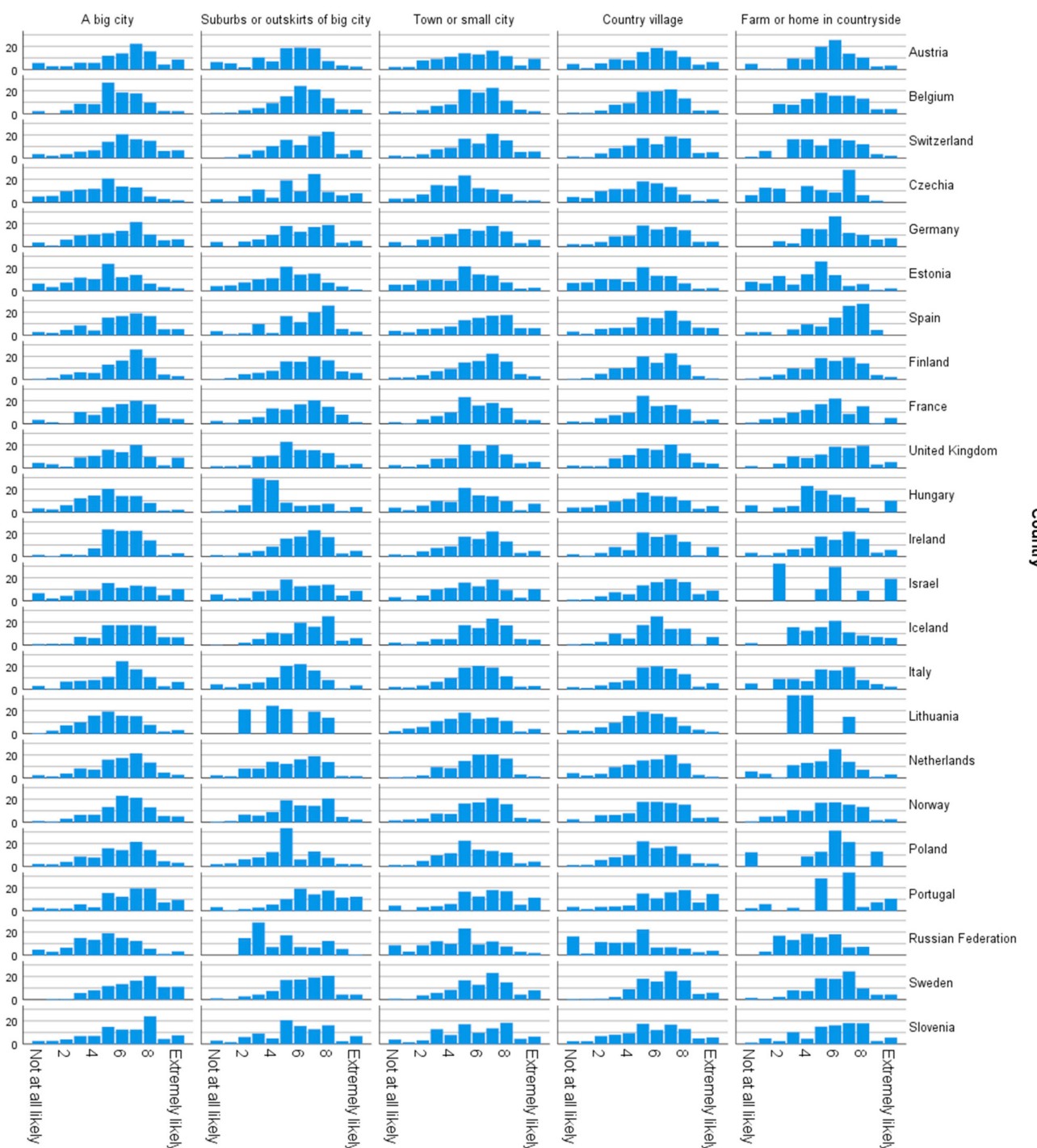

**Figure A4.** Expectation for positive outcomes from collective contribution by limiting own energy use (EES8 survey question: Now imagine that large numbers of people limited their energy use. How likely do you think it is that this would reduce climate change?); percentage data distributions broken down by country and domicile categories.

## Appendix B.

**Table A1.** Descriptive Statistics of Measures of Belief in Personal and Collective Ability to Limit Energy Use and Positive Climate Change Mitigation Outcomes Expectations (Big Cities—C, Suburbs—S, Town—T, Villages—V, Farms—F).

| | | How Confident You Could Use Less Energy Than Now | | | | | How Likely, Limiting Own Energy Use Reduce Climate Change | | | | | How Likely, Large Numbers of People Limit Energy Use | | | | | Imagine Large Numbers of People Limit Energy Use, How Likely Reduce Climate Change | | | | |
|---|---|---|---|---|---|---|---|---|---|---|---|---|---|---|---|---|---|---|---|---|---|
| | | C | S | T | V | F | C | S | T | V | F | C | S | T | V | F | C | S | T | V | F |
| Austria | N, valid | 471 | 141 | 475 | 778 | 110 | 469 | 137 | 455 | 734 | 103 | 468 | 137 | 461 | 741 | 108 | 465 | 139 | 461 | 747 | 107 |
| | N, missing | 9 | 2 | 4 | 18 | 1 | 11 | 5 | 24 | 63 | 9 | 12 | 6 | 19 | 56 | 4 | 15 | 3 | 19 | 50 | 4 |
| | Mean | 6.112 | 6.279 | 6.272 | 5.758 | 5.776 | 5.137 | 4.339 | 5.159 | 4.949 | 4.543 | 4.096 | 3.731 | 4.322 | 4.016 | 4.310 | 6.006 | 5.170 | 5.674 | 5.621 | 5.542 |
| | Std. Error | 0.122 | 0.192 | 0.112 | 0.090 | 0.208 | 0.126 | 0.227 | 0.127 | 0.099 | 0.240 | 0.107 | 0.166 | 0.105 | 0.082 | 0.178 | 0.120 | 0.206 | 0.117 | 0.090 | 0.207 |
| | Std. Dev. | 2.651 | 2.276 | 2.444 | 2.520 | 2.183 | 2.720 | 2.658 | 2.711 | 2.685 | 2.437 | 2.307 | 1.946 | 2.258 | 2.228 | 1.843 | 2.597 | 2.428 | 2.512 | 2.460 | 2.147 |
| Belgium | N, valid | 301 | 155 | 404 | 811 | 93 | 296 | 151 | 403 | 803 | 92 | 297 | 151 | 402 | 803 | 92 | 297 | 150 | 402 | 802 | 92 |
| | N, missing | 0 | 0 | 0 | 1 | 0 | 5 | 4 | 1 | 10 | 1 | 4 | 4 | 3 | 10 | 1 | 4 | 5 | 2 | 10 | 1 |
| | Mean | 6.346 | 6.595 | 6.752 | 6.581 | 6.164 | 4.703 | 5.035 | 4.808 | 4.940 | 4.893 | 4.514 | 4.234 | 4.189 | 4.405 | 4.292 | 5.578 | 6.033 | 5.778 | 5.863 | 5.690 |
| | Std. Error | 0.130 | 0.172 | 0.102 | 0.079 | 0.299 | 0.143 | 0.199 | 0.118 | 0.081 | 0.261 | 0.102 | 0.159 | 0.091 | 0.064 | 0.176 | 0.112 | 0.157 | 0.099 | 0.068 | 0.219 |
| | Std. Dev. | 2.256 | 2.148 | 2.059 | 2.242 | 2.893 | 2.459 | 2.449 | 2.376 | 2.291 | 2.507 | 1.758 | 1.948 | 1.819 | 1.809 | 1.694 | 1.924 | 1.919 | 1.978 | 1.915 | 2.101 |
| Switzerland | N, valid | 121 | 117 | 424 | 780 | 63 | 116 | 112 | 411 | 772 | 63 | 116 | 116 | 413 | 773 | 64 | 115 | 114 | 414 | 762 | 63 |
| | N, missing | 0 | 0 | 7 | 13 | 1 | 5 | 4 | 20 | 21 | 1 | 5 | 1 | 18 | 19 | 0 | 7 | 2 | 17 | 30 | 1 |
| | Mean | 6.874 | 6.900 | 6.770 | 6.861 | 6.690 | 5.240 | 4.559 | 5.022 | 4.934 | 4.836 | 4.048 | 3.776 | 4.052 | 3.998 | 3.901 | 6.032 | 6.301 | 5.993 | 5.927 | 5.270 |
| | Std. Error | 0.188 | 0.200 | 0.109 | 0.073 | 0.247 | 0.253 | 0.257 | 0.131 | 0.093 | 0.327 | 0.200 | 0.189 | 0.102 | 0.072 | 0.248 | 0.221 | 0.194 | 0.111 | 0.079 | 0.283 |
| | Std. Dev. | 2.068 | 2.164 | 2.239 | 2.028 | 1.955 | 2.727 | 2.729 | 2.656 | 2.590 | 2.596 | 2.154 | 2.037 | 2.080 | 1.988 | 1.977 | 2.371 | 2.073 | 2.257 | 2.194 | 2.242 |

**Table A1.** *Cont.*

| | | How Confident You Could Use Less Energy Than Now | | | | | How Likely, Limiting Own Energy Use Reduce Climate Change | | | | | How Likely, Large Numbers of People Limit Energy Use | | | | | Imagine Large Numbers of People Limit Energy Use, How Likely Reduce Climate Change | | | | |
|---|---|---|---|---|---|---|---|---|---|---|---|---|---|---|---|---|---|---|---|---|---|
| | | C | S | T | V | F | C | S | T | V | F | C | S | T | V | F | C | S | T | V | F |
| Czechia | N, valid | 690 | 89 | 738 | 698 | 31 | 661 | 89 | 686 | 664 | 29 | 650 | 89 | 685 | 655 | 29 | 651 | 89 | 669 | 651 | 30 |
| | N, missing | 6 | 1 | 8 | 7 | 0 | 36 | 1 | 60 | 41 | 2 | 46 | 1 | 61 | 50 | 2 | 45 | 1 | 77 | 54 | 1 |
| | Mean | 5.324 | 5.236 | 5.362 | 5.100 | 5.734 | 3.454 | 3.946 | 3.585 | 3.381 | 4.531 | 3.395 | 3.757 | 3.680 | 3.513 | 4.116 | 4.644 | 5.902 | 4.768 | 4.809 | 4.580 |
| | Std. Error | 0.090 | 0.266 | 0.089 | 0.094 | 0.468 | 0.099 | 0.312 | 0.090 | 0.099 | 0.489 | 0.090 | 0.247 | 0.078 | 0.086 | 0.480 | 0.092 | 0.257 | 0.082 | 0.091 | 0.479 |
| | Std. Dev. | 2.359 | 2.507 | 2.425 | 2.477 | 2.617 | 2.535 | 2.942 | 2.358 | 2.559 | 2.638 | 2.287 | 2.322 | 2.053 | 2.211 | 2.589 | 2.343 | 2.419 | 2.134 | 2.318 | 2.619 |
| Germany | N, valid | 405 | 389 | 996 | 969 | 69 | 394 | 386 | 980 | 950 | 66 | 396 | 383 | 976 | 951 | 66 | 397 | 386 | 973 | 952 | 66 |
| | N, missing | 1 | 1 | 12 | 8 | 1 | 13 | 3 | 28 | 27 | 3 | 11 | 6 | 31 | 26 | 3 | 10 | 3 | 34 | 26 | 3 |
| | Mean | 6.697 | 6.343 | 6.227 | 6.359 | 6.195 | 4.445 | 4.524 | 4.357 | 4.429 | 4.921 | 3.626 | 3.726 | 3.769 | 3.783 | 4.260 | 5.722 | 5.857 | 5.617 | 5.715 | 6.066 |
| | Std. Error | 0.121 | 0.117 | 0.080 | 0.074 | 0.253 | 0.146 | 0.138 | 0.085 | 0.087 | 0.307 | 0.102 | 0.099 | 0.064 | 0.062 | 0.233 | 0.123 | 0.118 | 0.077 | 0.073 | 0.249 |
| | Std. Dev. | 2.440 | 2.300 | 2.518 | 2.311 | 2.103 | 2.891 | 2.720 | 2.661 | 2.694 | 2.497 | 2.037 | 1.939 | 2.005 | 1.906 | 1.890 | 2.451 | 2.321 | 2.413 | 2.253 | 2.023 |
| Estonia | N, valid | 639 | 190 | 614 | 447 | 118 | 621 | 185 | 590 | 433 | 114 | 616 | 187 | 586 | 428 | 114 | 614 | 184 | 584 | 419 | 114 |
| | N, missing | 1 | 0 | 6 | 4 | 1 | 19 | 5 | 30 | 18 | 4 | 23 | 4 | 35 | 22 | 5 | 26 | 6 | 36 | 31 | 5 |
| | Mean | 5.310 | 6.560 | 5.621 | 4.918 | 5.274 | 3.198 | 3.046 | 3.468 | 3.045 | 2.820 | 3.655 | 3.295 | 3.662 | 3.543 | 3.644 | 4.827 | 4.940 | 4.799 | 4.551 | 4.253 |
| | Std. Error | 0.118 | 0.180 | 0.119 | 0.141 | 0.272 | 0.102 | 0.184 | 0.107 | 0.124 | 0.222 | 0.081 | 0.152 | 0.090 | 0.111 | 0.211 | 0.096 | 0.171 | 0.100 | 0.122 | 0.222 |
| | Std. Dev. | 2.991 | 2.475 | 2.959 | 2.978 | 2.950 | 2.540 | 2.508 | 2.595 | 2.575 | 2.374 | 2.005 | 2.074 | 2.174 | 2.294 | 2.253 | 2.382 | 2.314 | 2.423 | 2.503 | 2.366 |

**Table A1.** *Cont.*

| | | How Confident You Could Use Less Energy Than Now | | | | | How Likely, Limiting Own Energy Use Reduce Climate Change | | | | | How Likely, Large Numbers of People Limit Energy Use | | | | | Imagine Large Numbers of People Limit Energy Use, How Likely Reduce Climate Change | | | | |
|---|---|---|---|---|---|---|---|---|---|---|---|---|---|---|---|---|---|---|---|---|---|
| | | **C** | **S** | **T** | **V** | **F** | **C** | **S** | **T** | **V** | **F** | **C** | **S** | **T** | **V** | **F** | **C** | **S** | **T** | **V** | **F** |
| Spain | N, valid | 379 | 113 | 510 | 816 | 44 | 371 | 109 | 509 | 796 | 39 | 363 | 111 | 498 | 761 | 40 | 360 | 109 | 483 | 746 | 34 |
| | N, missing | 21 | 2 | 22 | 43 | 4 | 29 | 6 | 23 | 63 | 8 | 37 | 5 | 34 | 99 | 8 | 40 | 7 | 49 | 113 | 13 |
| | Mean | 5.521 | 5.756 | 5.640 | 5.387 | 5.464 | 4.865 | 4.670 | 4.891 | 4.625 | 4.714 | 4.018 | 3.883 | 4.069 | 4.056 | 3.603 | 5.944 | 6.176 | 5.914 | 5.956 | 6.222 |
| | Std. Error | 0.132 | 0.208 | 0.114 | 0.090 | 0.394 | 0.140 | 0.230 | 0.119 | 0.091 | 0.429 | 0.123 | 0.205 | 0.107 | 0.085 | 0.370 | 0.124 | 0.217 | 0.114 | 0.087 | 0.357 |
| | Std. Dev. | 2.575 | 2.216 | 2.565 | 2.577 | 2.605 | 2.700 | 2.402 | 2.676 | 2.566 | 2.679 | 2.345 | 2.157 | 2.380 | 2.340 | 2.327 | 2.349 | 2.265 | 2.498 | 2.378 | 2.093 |
| Finland | N, valid | 425 | 247 | 524 | 341 | 374 | 423 | 244 | 516 | 335 | 372 | 423 | 243 | 518 | 338 | 372 | 424 | 243 | 522 | 334 | 371 |
| | N, missing | 2 | 1 | 3 | 1 | 4 | 4 | 4 | 11 | 7 | 7 | 4 | 5 | 10 | 5 | 6 | 3 | 5 | 6 | 8 | 7 |
| | Mean | 6.845 | 6.974 | 6.919 | 6.652 | 6.517 | 4.063 | 4.390 | 4.156 | 4.110 | 3.873 | 4.231 | 4.366 | 4.384 | 4.549 | 4.384 | 6.178 | 6.212 | 5.896 | 5.613 | 5.684 |
| | Std. Error | 0.110 | 0.134 | 0.093 | 0.114 | 0.116 | 0.127 | 0.166 | 0.116 | 0.136 | 0.128 | 0.100 | 0.141 | 0.090 | 0.117 | 0.107 | 0.098 | 0.137 | 0.094 | 0.106 | 0.108 |
| | Std. Dev. | 2.259 | 2.105 | 2.120 | 2.110 | 2.246 | 2.613 | 2.596 | 2.634 | 2.485 | 2.470 | 2.056 | 2.196 | 2.055 | 2.141 | 2.069 | 2.015 | 2.136 | 2.137 | 1.946 | 2.086 |
| France | N, valid | 345 | 245 | 724 | 612 | 134 | 344 | 244 | 712 | 595 | 134 | 345 | 243 | 714 | 594 | 134 | 344 | 242 | 710 | 593 | 130 |
| | N, missing | 1 | 0 | 3 | 4 | 1 | 2 | 1 | 15 | 22 | 2 | 1 | 2 | 13 | 22 | 2 | 2 | 3 | 18 | 24 | 5 |
| | Mean | 7.377 | 7.250 | 7.328 | 7.149 | 7.285 | 4.909 | 4.939 | 4.897 | 4.382 | 4.367 | 4.266 | 4.087 | 4.147 | 3.863 | 4.084 | 5.966 | 5.885 | 5.797 | 5.593 | 5.443 |
| | Std. Error | 0.100 | 0.115 | 0.068 | 0.086 | 0.161 | 0.147 | 0.161 | 0.089 | 0.104 | 0.220 | 0.110 | 0.114 | 0.067 | 0.073 | 0.167 | 0.120 | 0.139 | 0.075 | 0.087 | 0.196 |
| | Std. Dev. | 1.863 | 1.796 | 1.818 | 2.129 | 1.862 | 2.736 | 2.507 | 2.388 | 2.535 | 2.549 | 2.045 | 1.769 | 1.800 | 1.788 | 1.936 | 2.226 | 2.157 | 2.000 | 2.117 | 2.236 |

**Table A1.** *Cont.*

| | | How Confident You Could Use Less Energy Than Now | | | | | How Likely, Limiting Own Energy Use Reduce Climate Change | | | | | How Likely, Large Numbers of People Limit Energy Use | | | | | Imagine Large Numbers of People Limit Energy Use, How Likely Reduce Climate Change | | | | |
|---|---|---|---|---|---|---|---|---|---|---|---|---|---|---|---|---|---|---|---|---|
| | | **C** | **S** | **T** | **V** | **F** | **C** | **S** | **T** | **V** | **F** | **C** | **S** | **T** | **V** | **F** | **C** | **S** | **T** | **V** | **F** |
| United Kingdom | N, valid | 222 | 384 | 859 | 391 | 94 | 217 | 371 | 838 | 382 | 94 | 216 | 371 | 832 | 383 | 94 | 218 | 368 | 828 | 380 | 93 |
| | N, missing | 1 | 1 | 5 | 0 | 0 | 7 | 14 | 26 | 8 | 0 | 8 | 13 | 31 | 8 | 0 | 6 | 17 | 35 | 10 | 1 |
| | Mean | 6.485 | 6.332 | 6.499 | 6.377 | 5.839 | 4.377 | 4.566 | 4.403 | 4.222 | 4.040 | 3.652 | 3.721 | 3.762 | 3.853 | 3.731 | 5.687 | 5.616 | 5.824 | 5.823 | 6.006 |
| | Std. Error | 0.175 | 0.131 | 0.081 | 0.135 | 0.271 | 0.170 | 0.134 | 0.086 | 0.127 | 0.239 | 0.124 | 0.096 | 0.071 | 0.098 | 0.183 | 0.170 | 0.109 | 0.077 | 0.110 | 0.226 |
| | Std. Dev. | 2.607 | 2.570 | 2.368 | 2.673 | 2.629 | 2.511 | 2.576 | 2.488 | 2.478 | 2.318 | 1.817 | 1.849 | 2.052 | 1.917 | 1.775 | 2.505 | 2.087 | 2.207 | 2.149 | 2.181 |
| Hungary | N, valid | 402 | 88 | 567 | 502 | 15 | 392 | 91 | 556 | 480 | 11 | 401 | 91 | 533 | 474 | 11 | 392 | 89 | 539 | 471 | 12 |
| | N, missing | 11 | 5 | 10 | 13 | 0 | 22 | 2 | 22 | 35 | 4 | 12 | 2 | 44 | 41 | 4 | 21 | 4 | 38 | 44 | 3 |
| | Mean | 5.368 | 4.749 | 5.066 | 4.752 | 4.660 | 4.680 | 3.953 | 4.285 | 4.313 | 4.145 | 3.859 | 3.434 | 3.749 | 3.655 | 3.863 | 4.999 | 4.499 | 5.420 | 5.243 | 5.265 |
| | Std. Error | 0.128 | 0.211 | 0.111 | 0.123 | 0.888 | 0.128 | 0.231 | 0.107 | 0.125 | 0.887 | 0.111 | 0.182 | 0.092 | 0.109 | 0.698 | 0.110 | 0.229 | 0.105 | 0.115 | 0.736 |
| | Std. Dev. | 2.562 | 1.976 | 2.643 | 2.762 | 3.420 | 2.527 | 2.204 | 2.522 | 2.735 | 2.882 | 2.229 | 1.735 | 2.114 | 2.378 | 2.300 | 2.184 | 2.165 | 2.440 | 2.501 | 2.522 |
| Ireland | N, valid | 215 | 609 | 799 | 344 | 769 | 213 | 610 | 783 | 337 | 753 | 213 | 599 | 774 | 335 | 750 | 215 | 604 | 764 | 327 | 730 |
| | N, missing | 1 | 5 | 2 | 1 | 5 | 3 | 4 | 18 | 8 | 21 | 4 | 15 | 26 | 11 | 24 | 2 | 10 | 37 | 19 | 43 |
| | Mean | 6.651 | 7.029 | 6.230 | 6.602 | 6.267 | 5.892 | 4.800 | 4.879 | 4.751 | 4.328 | 5.384 | 4.481 | 4.337 | 4.205 | 3.967 | 6.038 | 6.108 | 5.842 | 5.946 | 5.958 |
| | Std. Error | 0.112 | 0.083 | 0.086 | 0.139 | 0.092 | 0.130 | 0.101 | 0.087 | 0.140 | 0.093 | 0.133 | 0.089 | 0.079 | 0.122 | 0.079 | 0.121 | 0.083 | 0.079 | 0.122 | 0.084 |
| | Std. Dev. | 1.642 | 2.057 | 2.422 | 2.580 | 2.543 | 1.900 | 2.507 | 2.426 | 2.565 | 2.542 | 1.942 | 2.169 | 2.203 | 2.233 | 2.150 | 1.772 | 2.042 | 2.179 | 2.199 | 2.271 |

**Table A1.** *Cont.*

| | | How Confident You Could Use Less Energy Than Now | | | | | How Likely, Limiting Own Energy Use Reduce Climate Change | | | | | How Likely, Large Numbers of People Limit Energy Use | | | | | Imagine Large Numbers of People Limit Energy Use, How Likely Reduce Climate Change | | | | |
|---|---|---|---|---|---|---|---|---|---|---|---|---|---|---|---|---|---|---|---|---|---|
| | | C | S | T | V | F | C | S | T | V | F | C | S | T | V | F | C | S | T | V | F |
| Israel | N, valid | 1426 | 278 | 365 | 297 | 6 | 1216 | 231 | 319 | 276 | 5 | 1181 | 231 | 310 | 273 | 5 | 1177 | 236 | 296 | 267 | 6 |
| | N, missing | 112 | 12 | 40 | 20 | 0 | 322 | 59 | 86 | 42 | 1 | 357 | 59 | 96 | 44 | 1 | 361 | 54 | 109 | 51 | 0 |
| | Mean | 5.707 | 5.806 | 5.326 | 5.995 | 6.537 | 3.898 | 4.634 | 4.636 | 5.165 | 3.462 | 4.182 | 4.614 | 4.333 | 4.227 | 3.854 | 5.632 | 5.740 | 5.747 | 6.275 | 5.550 |
| | Std. Error | 0.075 | 0.154 | 0.147 | 0.148 | 1.133 | 0.085 | 0.181 | 0.156 | 0.153 | 0.761 | 0.074 | 0.149 | 0.126 | 0.131 | 1.420 | 0.080 | 0.169 | 0.144 | 0.140 | 1.278 |
| | Std. Dev. | 2.831 | 2.572 | 2.818 | 2.558 | 2.823 | 2.959 | 2.758 | 2.784 | 2.541 | 1.667 | 2.541 | 2.264 | 2.216 | 2.164 | 3.113 | 2.761 | 2.593 | 2.469 | 2.284 | 3.183 |
| Iceland | N, valid | 120 | 215 | 435 | 75 | 26 | 119 | 210 | 432 | 75 | 26 | 119 | 210 | 429 | 75 | 27 | 118 | 210 | 425 | 75 | 27 |
| | N, missing | 2 | 0 | 3 | 1 | 0 | 4 | 5 | 6 | 1 | 0 | 3 | 5 | 9 | 1 | 0 | 4 | 5 | 12 | 1 | 0 |
| | Mean | 7.399 | 6.939 | 7.142 | 7.006 | 5.407 | 4.485 | 4.866 | 4.626 | 4.250 | 5.274 | 3.835 | 3.783 | 3.927 | 3.796 | 4.410 | 6.303 | 6.437 | 6.178 | 5.910 | 5.754 |
| | Std. Error | 0.218 | 0.176 | 0.121 | 0.303 | 0.523 | 0.259 | 0.182 | 0.134 | 0.318 | 0.564 | 0.184 | 0.114 | 0.090 | 0.187 | 0.452 | 0.195 | 0.136 | 0.104 | 0.243 | 0.432 |
| | Std. Dev. | 2.385 | 2.579 | 2.523 | 2.628 | 2.687 | 2.823 | 2.637 | 2.781 | 2.747 | 2.902 | 2.011 | 1.656 | 1.873 | 1.621 | 2.342 | 2.119 | 1.965 | 2.153 | 2.098 | 2.235 |
| Italy | N, valid | 279 | 145 | 866 | 1116 | 100 | 275 | 144 | 861 | 1080 | 95 | 263 | 145 | 852 | 1072 | 94 | 268 | 143 | 843 | 1080 | 94 |
| | N, missing | 17 | 8 | 49 | 42 | 2 | 21 | 8 | 54 | 78 | 7 | 33 | 8 | 62 | 85 | 8 | 28 | 10 | 72 | 78 | 7 |
| | Mean | 6.143 | 5.635 | 6.059 | 6.073 | 5.966 | 4.899 | 4.667 | 4.659 | 4.974 | 4.059 | 4.548 | 4.616 | 4.411 | 4.618 | 4.598 | 5.734 | 5.332 | 5.655 | 5.863 | 5.326 |
| | Std. Error | 0.140 | 0.209 | 0.079 | 0.066 | 0.253 | 0.144 | 0.189 | 0.080 | 0.075 | 0.251 | 0.149 | 0.190 | 0.074 | 0.069 | 0.222 | 0.142 | 0.184 | 0.071 | 0.065 | 0.244 |
| | Std. Dev. | 2.331 | 2.512 | 2.324 | 2.222 | 2.534 | 2.388 | 2.270 | 2.336 | 2.479 | 2.439 | 2.409 | 2.281 | 2.162 | 2.265 | 2.153 | 2.326 | 2.201 | 2.071 | 2.137 | 2.367 |

**Table A1.** *Cont.*

| | | How Confident You Could Use Less Energy Than Now | | | | | How Likely, Limiting Own Energy Use Reduce Climate Change | | | | | How Likely, Large Numbers of People Limit Energy Use | | | | | Imagine Large Numbers of People Limit Energy Use, How Likely Reduce Climate Change | | | | |
|---|---|---|---|---|---|---|---|---|---|---|---|---|---|---|---|---|---|---|---|---|---|
| | | C | S | T | V | F | C | S | T | V | F | C | S | T | V | F | C | S | T | V | F |
| Lithuania | N, valid | 788 | 7 | 679 | 555 | 6 | 695 | 6 | 621 | 493 | 6 | 699 | 6 | 612 | 481 | 6 | 682 | 7 | 606 | 483 | 6 |
| | N, missing | 26 | 2 | 35 | 23 | 0 | 118 | 3 | 93 | 86 | 0 | 114 | 3 | 102 | 98 | 0 | 131 | 2 | 108 | 96 | 0 |
| | Mean | 5.486 | 5.613 | 6.066 | 5.355 | 6.217 | 4.853 | 5.371 | 4.989 | 4.772 | 6.162 | 4.324 | 4.514 | 4.531 | 4.625 | 4.757 | 5.223 | 4.923 | 5.210 | 5.147 | 3.992 |
| | Std. Error | 0.088 | 1.354 | 0.095 | 0.111 | 0.701 | 0.093 | 1.160 | 0.106 | 0.118 | 0.778 | 0.078 | 0.997 | 0.097 | 0.103 | 0.877 | 0.080 | 0.849 | 0.095 | 0.098 | 0.570 |
| | Std. Dev. | 2.480 | 3.681 | 2.472 | 2.624 | 1.784 | 2.452 | 2.917 | 2.640 | 2.619 | 1.979 | 2.053 | 2.507 | 2.403 | 2.257 | 2.230 | 2.099 | 2.210 | 2.341 | 2.147 | 1.451 |
| Nether-lands | N, valid | 326 | 143 | 438 | 676 | 86 | 317 | 139 | 436 | 667 | 83 | 314 | 140 | 436 | 666 | 84 | 312 | 140 | 436 | 664 | 83 |
| | N, missing | 2 | 0 | 5 | 5 | 0 | 11 | 4 | 7 | 14 | 3 | 14 | 3 | 7 | 15 | 2 | 16 | 3 | 7 | 16 | 3 |
| | Mean | 6.337 | 6.324 | 6.483 | 6.379 | 6.592 | 4.498 | 3.979 | 4.524 | 4.177 | 4.373 | 4.253 | 4.213 | 4.372 | 4.397 | 4.547 | 5.790 | 5.370 | 5.895 | 5.391 | 5.150 |
| | Std. Error | 0.149 | 0.202 | 0.119 | 0.091 | 0.258 | 0.152 | 0.218 | 0.112 | 0.094 | 0.274 | 0.116 | 0.159 | 0.081 | 0.070 | 0.217 | 0.124 | 0.187 | 0.093 | 0.087 | 0.251 |
| | Std. Dev. | 2.699 | 2.422 | 2.494 | 2.370 | 2.397 | 2.703 | 2.564 | 2.346 | 2.430 | 2.500 | 2.060 | 1.874 | 1.692 | 1.815 | 1.986 | 2.191 | 2.215 | 1.934 | 2.242 | 2.280 |
| Norway | N, valid | 234 | 244 | 452 | 318 | 294 | 234 | 241 | 445 | 308 | 284 | 233 | 242 | 445 | 309 | 286 | 232 | 240 | 445 | 309 | 288 |
| | N, missing | 0 | 0 | 1 | 0 | 0 | 1 | 3 | 8 | 10 | 9 | 1 | 2 | 8 | 9 | 7 | 2 | 4 | 8 | 9 | 5 |
| | Mean | 7.259 | 7.383 | 7.279 | 7.087 | 7.178 | 4.358 | 4.089 | 4.163 | 3.926 | 3.754 | 4.607 | 4.700 | 4.571 | 4.497 | 4.510 | 6.136 | 5.875 | 5.850 | 5.797 | 5.380 |
| | Std. Error | 0.155 | 0.150 | 0.103 | 0.134 | 0.138 | 0.169 | 0.169 | 0.121 | 0.145 | 0.148 | 0.136 | 0.132 | 0.089 | 0.109 | 0.125 | 0.135 | 0.138 | 0.102 | 0.127 | 0.131 |
| | Std. Dev. | 2.366 | 2.346 | 2.201 | 2.381 | 2.356 | 2.581 | 2.619 | 2.560 | 2.554 | 2.498 | 2.085 | 2.046 | 1.874 | 1.915 | 2.108 | 2.059 | 2.136 | 2.143 | 2.231 | 2.224 |

**Table A1.** *Cont.*

| | | How Confident You Could Use Less Energy Than Now | | | | | How Likely, Limiting Own Energy Use Reduce Climate Change | | | | | How Likely, Large Numbers of People Limit Energy Use | | | | | Imagine Large Numbers of People Limit Energy Use, How Likely Reduce Climate Change | | | | |
|---|---|---|---|---|---|---|---|---|---|---|---|---|---|---|---|---|---|---|---|---|---|
| | | C | S | T | V | F | C | S | T | V | F | C | S | T | V | F | C | S | T | V | F |
| Poland | N, valid | 353 | 46 | 512 | 695 | 16 | 329 | 41 | 470 | 668 | 9 | 335 | 40 | 474 | 667 | 10 | 335 | 39 | 466 | 654 | 10 |
| | N, missing | 10 | 1 | 14 | 38 | 1 | 34 | 6 | 56 | 66 | 8 | 28 | 7 | 52 | 67 | 7 | 28 | 8 | 60 | 79 | 8 |
| | Mean | 5.893 | 5.877 | 5.741 | 5.548 | 4.145 | 4.046 | 3.103 | 4.028 | 4.392 | 4.761 | 3.890 | 3.648 | 3.834 | 4.193 | 3.388 | 5.785 | 5.058 | 5.521 | 5.514 | 5.548 |
| | Std. Error | 0.140 | 0.416 | 0.115 | 0.096 | 0.745 | 0.147 | 0.362 | 0.113 | 0.095 | 0.875 | 0.109 | 0.323 | 0.092 | 0.083 | 0.558 | 0.123 | 0.335 | 0.101 | 0.082 | 0.852 |
| | Std. Dev. | 2.632 | 2.830 | 2.607 | 2.544 | 2.996 | 2.664 | 2.314 | 2.447 | 2.451 | 2.692 | 1.995 | 2.036 | 1.994 | 2.139 | 1.790 | 2.255 | 2.087 | 2.170 | 2.092 | 2.629 |
| Portugal | N, valid | 212 | 164 | 389 | 467 | 28 | 207 | 158 | 377 | 438 | 26 | 204 | 158 | 370 | 431 | 25 | 200 | 158 | 373 | 426 | 25 |
| | N, missing | 1 | 0 | 2 | 4 | 2 | 6 | 5 | 14 | 34 | 4 | 9 | 5 | 21 | 41 | 4 | 13 | 6 | 18 | 45 | 4 |
| | Mean | 5.889 | 5.828 | 5.169 | 5.046 | 3.447 | 4.969 | 4.816 | 4.379 | 4.385 | 3.507 | 3.800 | 4.023 | 3.852 | 4.210 | 3.876 | 6.438 | 6.739 | 6.306 | 6.527 | 6.294 |
| | Std. Error | 0.208 | 0.230 | 0.159 | 0.151 | 0.504 | 0.206 | 0.229 | 0.153 | 0.145 | 0.627 | 0.169 | 0.180 | 0.127 | 0.136 | 0.393 | 0.169 | 0.188 | 0.129 | 0.125 | 0.489 |
| | Std. Dev. | 3.023 | 2.948 | 3.144 | 3.271 | 2.649 | 2.961 | 2.878 | 2.963 | 3.044 | 3.180 | 2.416 | 2.268 | 2.443 | 2.832 | 1.971 | 2.391 | 2.364 | 2.483 | 2.571 | 2.450 |
| Russian federation | N, valid | 814 | 72 | 767 | 569 | 29 | 714 | 73 | 626 | 465 | 28 | 709 | 72 | 644 | 432 | 28 | 695 | 73 | 635 | 420 | 28 |
| | N, missing | 55 | 5 | 52 | 54 | 0 | 155 | 4 | 194 | 158 | 1 | 159 | 5 | 175 | 190 | 1 | 174 | 4 | 184 | 203 | 1 |
| | Mean | 4.666 | 3.447 | 4.521 | 4.257 | 6.055 | 4.014 | 2.547 | 3.904 | 3.003 | 3.772 | 4.334 | 2.982 | 4.136 | 3.648 | 3.640 | 4.786 | 4.743 | 4.619 | 4.141 | 4.453 |
| | Std. Error | 0.090 | 0.356 | 0.095 | 0.110 | 0.329 | 0.086 | 0.249 | 0.104 | 0.109 | 0.365 | 0.085 | 0.270 | 0.098 | 0.116 | 0.318 | 0.087 | 0.263 | 0.099 | 0.134 | 0.363 |
| | Std. Dev. | 2.566 | 3.022 | 2.637 | 2.629 | 1.783 | 2.288 | 2.125 | 2.612 | 2.358 | 1.935 | 2.259 | 2.289 | 2.484 | 2.416 | 1.685 | 2.284 | 2.243 | 2.487 | 2.738 | 1.924 |

**Table A1.** *Cont.*

| | | How Confident You Could Use Less Energy Than Now | | | | | How Likely, Limiting Own Energy Use Reduce Climate Change | | | | | How Likely, Large Numbers of People Limit Energy Use | | | | | Imagine Large Numbers of People Limit Energy Use, How Likely Reduce Climate Change | | | | |
|---|---|---|---|---|---|---|---|---|---|---|---|---|---|---|---|---|---|---|---|---|---|
| | | **C** | **S** | **T** | **V** | **F** | **C** | **S** | **T** | **V** | **F** | **C** | **S** | **T** | **V** | **F** | **C** | **S** | **T** | **V** | **F** |
| Sweden | N, valid | 191 | 333 | 591 | 246 | 171 | 189 | 330 | 584 | 244 | 164 | 187 | 331 | 587 | 242 | 167 | 186 | 332 | 587 | 243 | 162 |
| | N, missing | 3 | 5 | 4 | 3 | 0 | 4 | 8 | 11 | 4 | 7 | 6 | 7 | 8 | 7 | 4 | 7 | 6 | 8 | 6 | 8 |
| | Mean | 7.899 | 7.217 | 7.300 | 6.884 | 6.644 | 4.582 | 4.458 | 4.559 | 4.257 | 4.227 | 4.580 | 4.727 | 4.717 | 4.729 | 4.830 | 6.833 | 6.250 | 6.239 | 6.411 | 5.943 |
| | Std. Error | 0.164 | 0.137 | 0.109 | 0.152 | 0.202 | 0.212 | 0.146 | 0.117 | 0.151 | 0.204 | 0.160 | 0.118 | 0.089 | 0.128 | 0.139 | 0.153 | 0.110 | 0.088 | 0.121 | 0.159 |
| | Std. Dev. | 2.267 | 2.502 | 2.657 | 2.378 | 2.644 | 2.908 | 2.652 | 2.832 | 2.354 | 2.607 | 2.189 | 2.143 | 2.161 | 1.989 | 1.800 | 2.091 | 2.000 | 2.142 | 1.882 | 2.026 |
| Slovenia | N, valid | 162 | 127 | 286 | 649 | 69 | 162 | 122 | 277 | 637 | 64 | 164 | 126 | 279 | 631 | 66 | 159 | 127 | 277 | 630 | 66 |
| | N, missing | 3 | 0 | 0 | 9 | 0 | 3 | 4 | 8 | 22 | 5 | 1 | 1 | 7 | 27 | 3 | 6 | 0 | 9 | 29 | 3 |
| | Mean | 5.900 | 5.481 | 5.944 | 5.786 | 6.003 | 3.215 | 3.387 | 3.690 | 3.420 | 3.566 | 3.744 | 3.593 | 3.636 | 3.548 | 3.329 | 6.078 | 5.711 | 5.735 | 5.638 | 5.860 |
| | Std. Error | 0.212 | 0.258 | 0.161 | 0.107 | 0.344 | 0.207 | 0.217 | 0.172 | 0.103 | 0.344 | 0.164 | 0.193 | 0.125 | 0.081 | 0.268 | 0.197 | 0.215 | 0.151 | 0.097 | 0.289 |
| | Std. Dev. | 2.690 | 2.909 | 2.728 | 2.737 | 2.861 | 2.630 | 2.404 | 2.858 | 2.596 | 2.758 | 2.101 | 2.166 | 2.090 | 2.026 | 2.184 | 2.479 | 2.417 | 2.507 | 2.426 | 2.354 |

**Appendix C. Sample Means, Sample Proportion of Answers Ranging from 6 to 10, and 95% Confidence Intervals for Population Mean and Proportion Evaluating Beliefs in Ability to Contribute to Climate Change Mitigation and Positive Outcomes Expectation in Different Countries by Domicile Categories Samples**

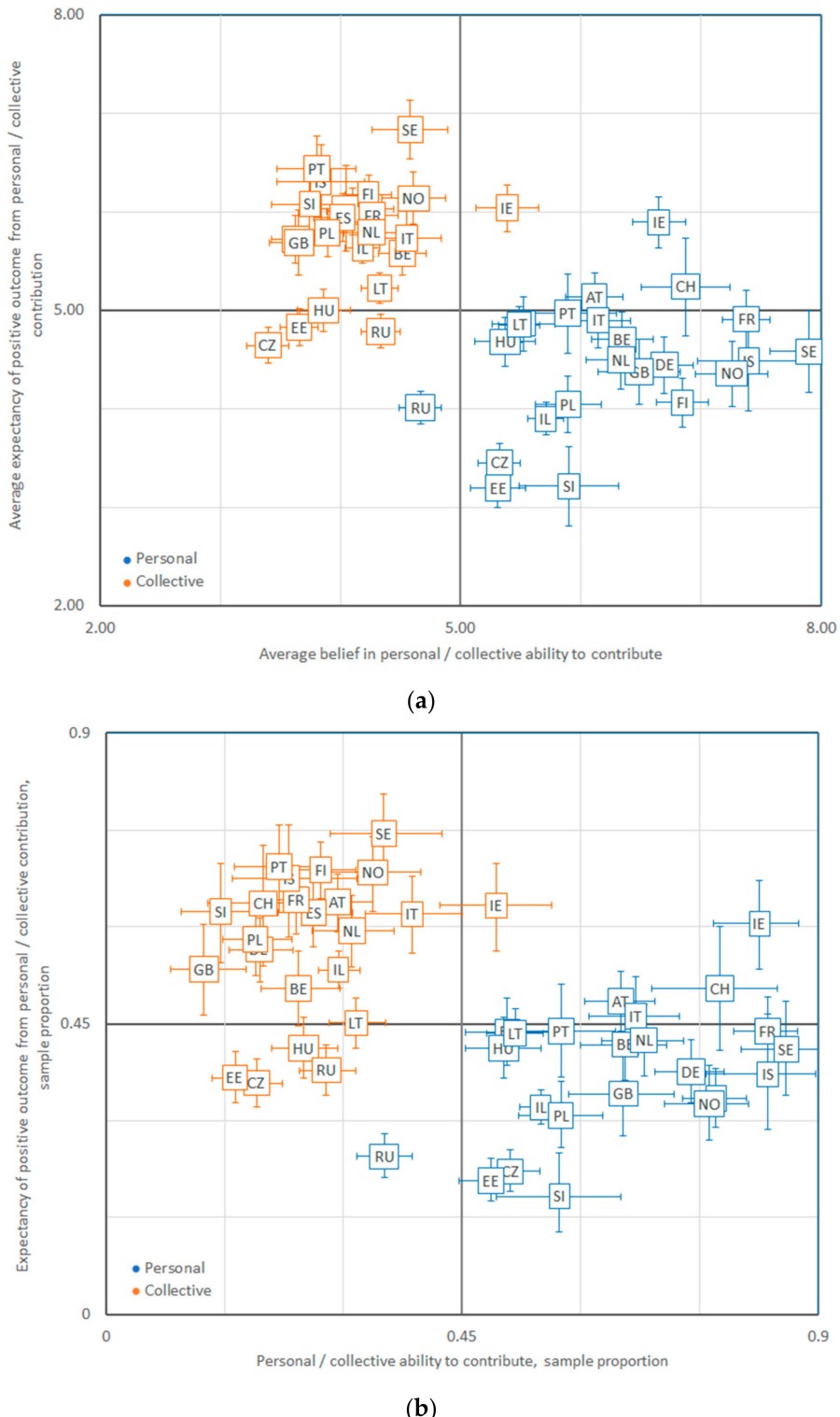

**Figure A5.** Big city samples: (**a**) mean and 95% confidence interval, (**b**) sample proportion and 95% Clopper–Pearson confidence interval.

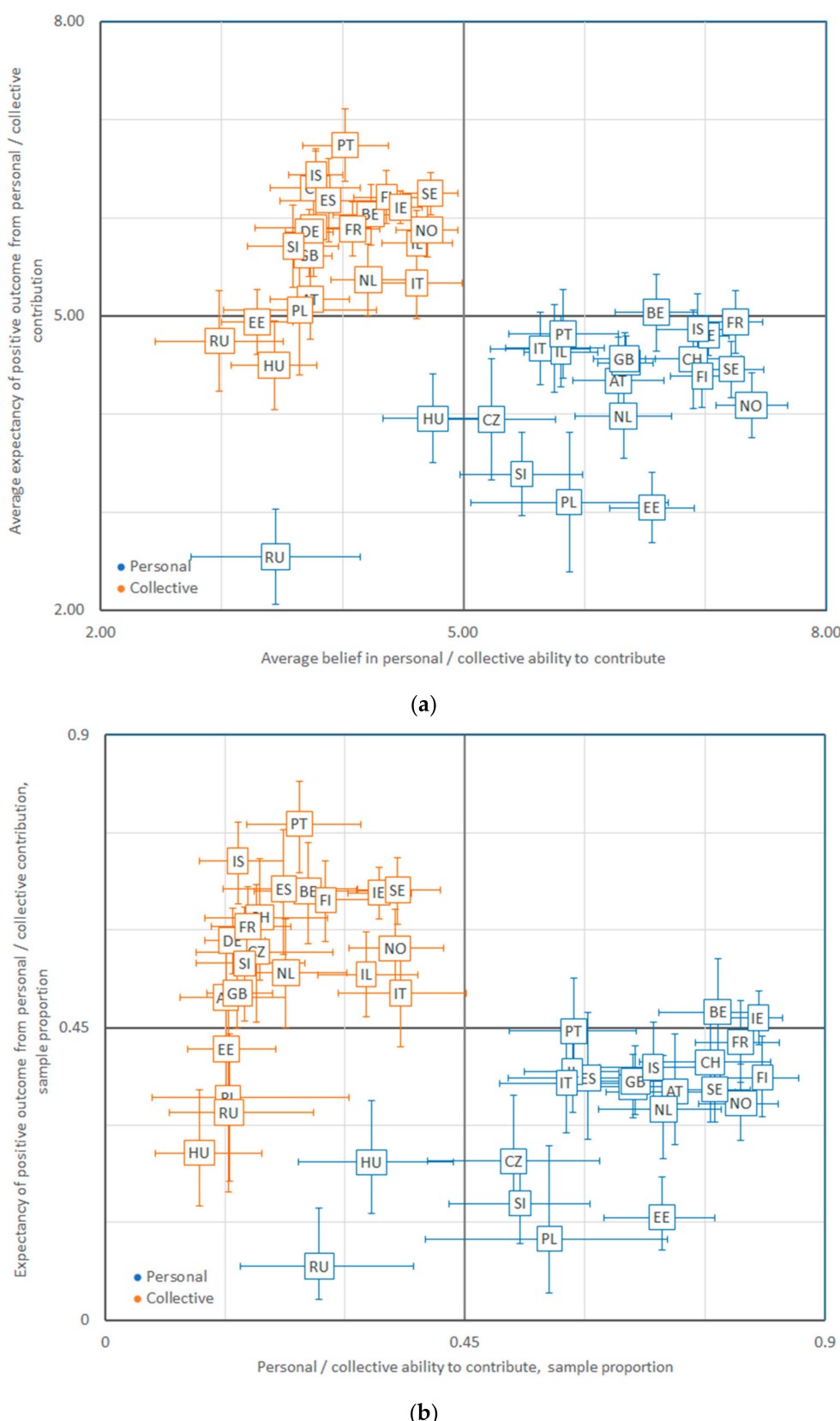

(**a**)

(**b**)

**Figure A6.** Suburbs and outskirts of big city samples: (**a**) mean and 95% confidence interval, (**b**) sample proportion and 95% Clopper–Pearson confidence interval.

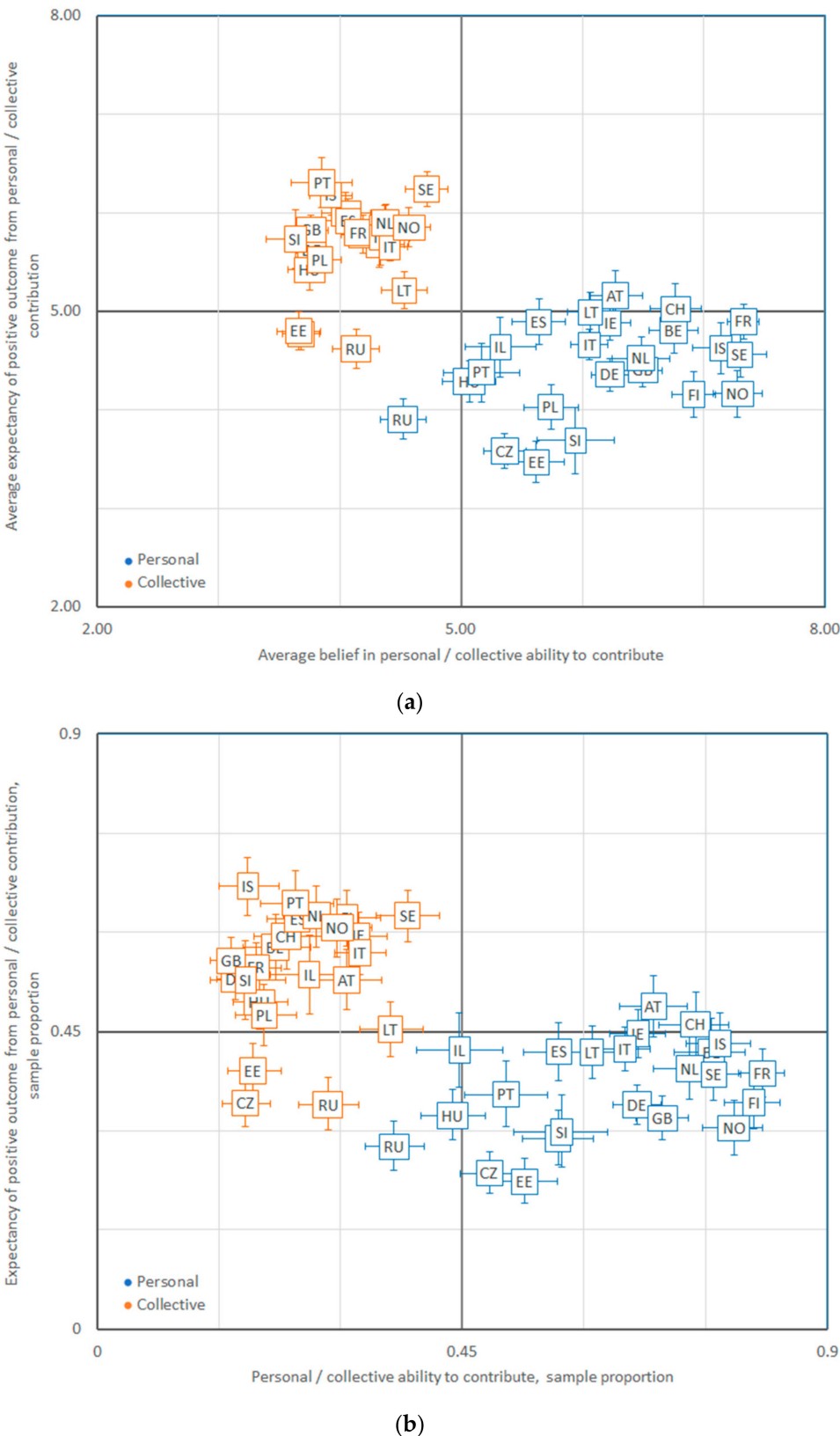

**Figure A7.** Town and small city samples: (**a**) mean and 95% confidence interval, (**b**) sample proportion and 95% Clopper–Pearson confidence interval.

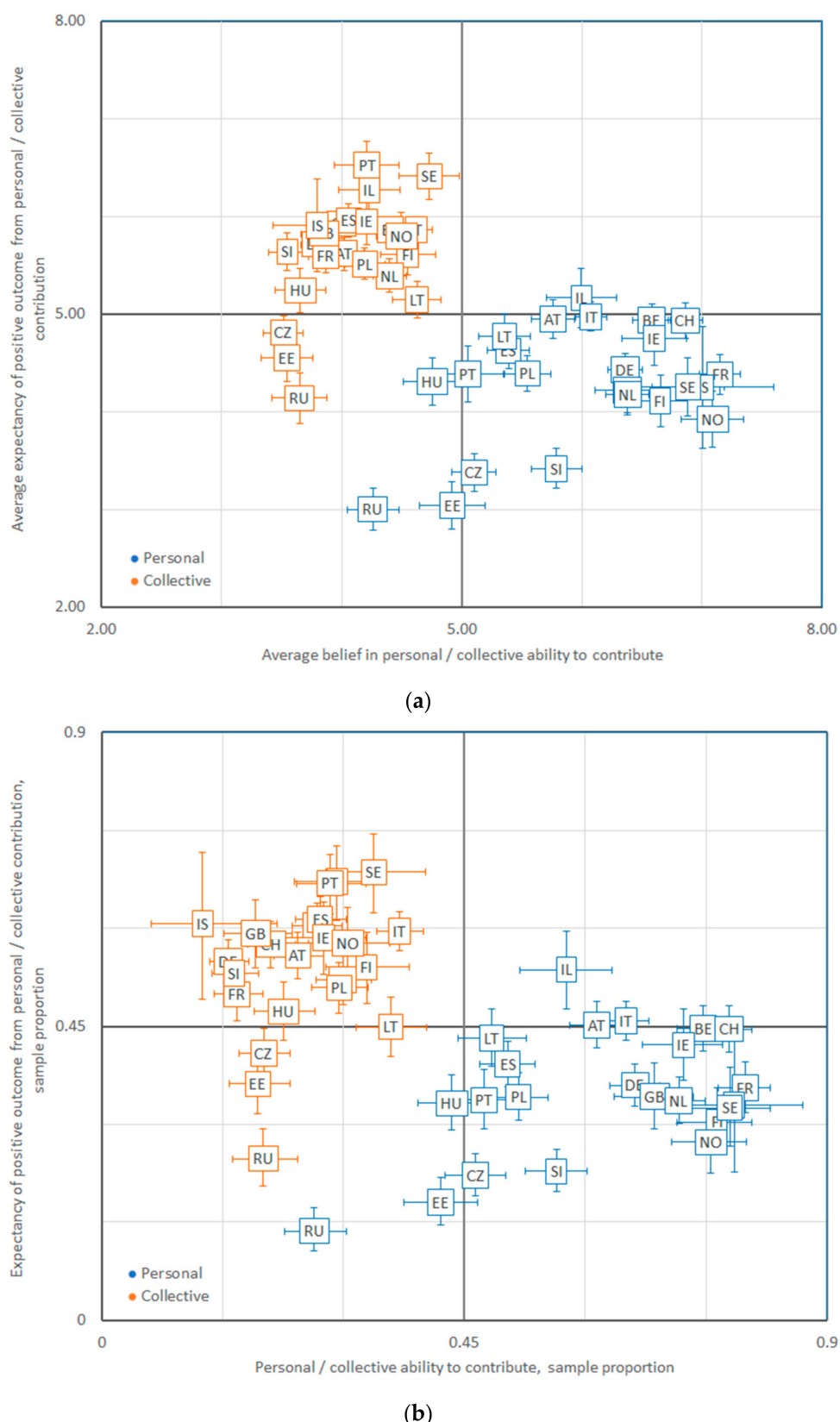

**Figure A8.** Country village samples: (**a**) mean and 95% confidence interval, (**b**) sample proportion and 95% Clopper–Pearson confidence interval.

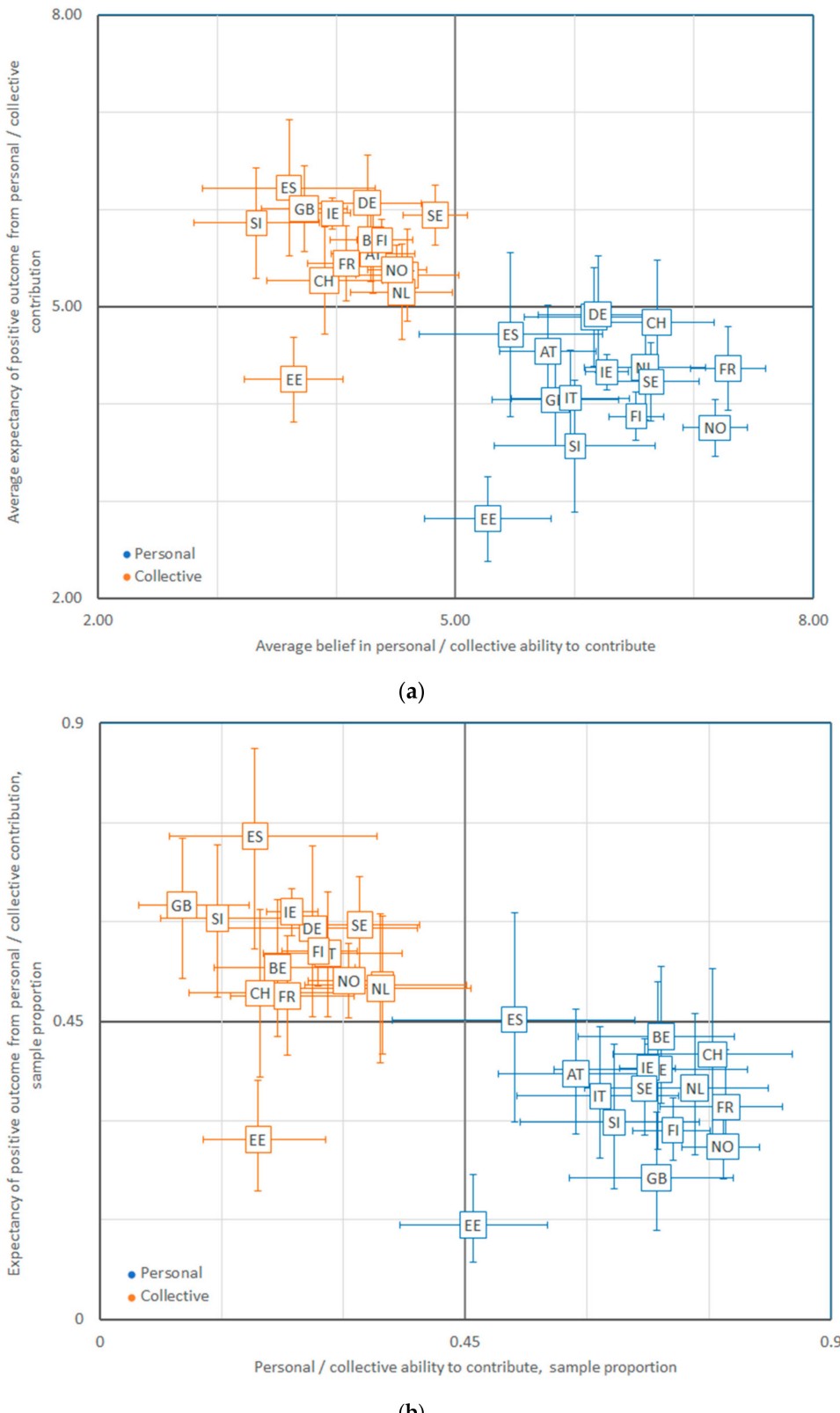

**Figure A9.** Farm of home in countryside samples: (**a**) mean and 95% confidence interval, (**b**) sample proportion and 95% Clopper–Pearson confidence interval.

## Appendix D

**Table A2.** Correlations between Beliefs in Personal and Collective Ability to Limit Energy Use and Positive Climate Change Mitigation Outcomes Expectations: Pearson Correlation Coefficients, Their Statistical Significance, and Pairwise Sample sizes (Big Cities—C, Suburbs—S, Town—T, Villages—V, Farms—F)

| | | Correlations between Beliefs in Personal Ability to Limit Own Energy Use and Climate Change Mitigation Outcomes Expectation Variables | | | | | Correlations between Beliefs in Collective Ability to Limit Energy Use and Climate Change Mitigation Outcomes Expectation Variables | | | | |
|---|---|---|---|---|---|---|---|---|---|---|---|
| | | **C** | **S** | **T** | **V** | **F** | **C** | **S** | **T** | **V** | **F** |
| AT, Austria | Pearson Corr. | 0.263 ** | 0.318 ** | 0.521 ** | 0.335 ** | 0.359 ** | 0.301 ** | 0.339 ** | 0.416 ** | 0.396 ** | 0.436 ** |
| | Sig. (2 tailed) | 0.000 | 0.000 | 0.000 | 0.000 | 0.000 | 0.000 | 0.000 | 0.000 | 0.000 | 0.000 |
| | N | 461 | 135 | 454 | 722 | 102 | 464 | 136 | 456 | 735 | 106 |
| BE, Belgium | Pearson Corr. | 0.284 ** | −0.042 | 0.085 | 0.243 ** | 0.189 | 0.295 ** | 0.324 ** | 0.300 ** | 0.302 ** | 0.323 ** |
| | Sig. (2 tailed) | 0.000 | 0.609 | 0.090 | 0.000 | 0.071 | 0.000 | 0.000 | 0.000 | 0.000 | 0.002 |
| | N | 296 | 151 | 403 | 802 | 92 | 297 | 150 | 401 | 801 | 92 |
| CH, Switzerland | Pearson Corr. | 0.280 ** | 0.346 ** | 0.210 ** | 0.279 ** | 0.409 ** | 0.247 ** | 0.237 * | 0.274 ** | 0.220 ** | 0.316 * |
| | Sig. (2 tailed) | 0.002 | 0.000 | 0.000 | 0.000 | 0.001 | 0.008 | 0.011 | 0.000 | 0.000 | 0.012 |
| | N | 116 | 112 | 407 | 759 | 62 | 114 | 114 | 412 | 762 | 63 |
| CZ, Czechia | Pearson Corr. | 0.210 ** | 0.395 ** | 0.189 ** | 0.245 ** | | 0.390 ** | 0.144 | 0.416 ** | 0.436 ** | |
| | Sig. (2 tailed) | 0.000 | 0.000 | 0.000 | 0.000 | | 0.000 | 0.179 | 0.000 | 0.000 | |
| | N | 655 | 87 | 681 | 658 | | 644 | 89 | 658 | 637 | |
| DE, Germany | Pearson Corr. | 0.235 ** | 0.266 ** | 0.153 ** | 0.234 ** | 0.297 * | 0.214 ** | 0.144 ** | 0.156 ** | 0.172 ** | 0.158 |
| | Sig. (2 tailed) | 0.000 | 0.000 | 0.000 | 0.000 | 0.015 | 0.000 | 0.005 | 0.000 | 0.000 | 0.205 |
| | N | 394 | 385 | 974 | 948 | 66 | 395 | 383 | 967 | 950 | 66 |
| EE, Estonia | Pearson Corr. | 0.305 ** | 0.253 ** | 0.290 ** | 0.330 ** | 0.159 | 0.330 ** | 0.400 ** | 0.452 ** | 0.492 ** | 0.497 ** |
| | Sig. (2 tailed) | 0.000 | 0.000 | 0.000 | 0.000 | 0.091 | 0.000 | 0.000 | 0.000 | 0.000 | 0.000 |
| | N | 620 | 185 | 587 | 431 | 114 | 613 | 184 | 581 | 418 | 113 |
| ES, Spain | Pearson Corr. | 0.278 ** | 0.037 | 0.278 ** | 0.263 ** | 0.400 * | 0.275 ** | 0.208 * | 0.357 ** | 0.305 ** | 0.270 |
| | Sig. (2 tailed) | 0.000 | 0.700 | 0.000 | 0.000 | 0.012 | 0.000 | 0.031 | 0.000 | 0.000 | 0.125 |
| | N | 362 | 109 | 492 | 772 | 39 | 349 | 108 | 473 | 720 | 33 |
| FI, Finland | Pearson Corr. | 0.223 ** | 0.175 ** | 0.281 ** | 0.159 ** | 0.305 ** | 0.239 ** | 0.320 ** | 0.260 ** | 0.280 ** | 0.282 ** |
| | Sig. (2 tailed) | 0.000 | 0.006 | 0.000 | 0.003 | 0.000 | 0.000 | 0.000 | 0.000 | 0.000 | 0.000 |
| | N | 422 | 244 | 516 | 335 | 369 | 423 | 243 | 518 | 334 | 369 |

**Table A2.** *Cont.*

| | | Correlations between Beliefs in Personal Ability to Limit Own Energy Use and Climate Change Mitigation Outcomes Expectation Variables | | | | | Correlations between Beliefs in Collective Ability to Limit Energy Use and Climate Change Mitigation Outcomes Expectation Variables | | | | |
|---|---|---|---|---|---|---|---|---|---|---|---|
| | | C | S | T | V | F | C | S | T | V | F |
| FR, France | Pearson Corr. | 0.245 ** | 0.181 ** | 0.176 ** | 0.191 ** | 0.022 | 0.310 ** | 0.321 ** | 0.280 ** | 0.325 ** | 0.378 ** |
| | Sig. (2 tailed) | 0.000 | 0.005 | 0.000 | 0.000 | 0.805 | 0.000 | 0.000 | 0.000 | 0.000 | 0.000 |
| | N | 344 | 244 | 710 | 595 | 133 | 344 | 242 | 708 | 590 | 130 |
| GB, United Kingdom | Pearson Corr. | 0.155 * | 0.202 ** | 0.152 ** | 0.039 | 0.114 | 0.189 ** | 0.237 ** | 0.222 ** | 0.223 ** | 0.178 |
| | Sig. (2 tailed) | 0.023 | 0.000 | 0.000 | 0.446 | 0.276 | 0.006 | 0.000 | 0.000 | 0.000 | 0.088 |
| | N | 216 | 370 | 833 | 382 | 94 | 213 | 367 | 826 | 380 | 93 |
| HU, Hungary | Pearson Corr. | 0.411 ** | 0.391 ** | 0.341 ** | 0.449 ** | | 0.262 ** | 0.272 * | 0.279 ** | 0.319 ** | |
| | Sig. (2 tailed) | 0.000 | 0.000 | 0.000 | 0.000 | | 0.000 | 0.010 | 0.000 | 0.000 | |
| | N | 384 | 86 | 548 | 472 | | 391 | 88 | 525 | 462 | |
| IE, Ireland | Pearson Corr. | 0.259 ** | −0.039 | 0.231 ** | 0.219 ** | 0.216 ** | 0.396 ** | 0.250 ** | 0.282 ** | 0.245 ** | 0.181 ** |
| | Sig. (2 tailed) | 0.000 | 0.342 | 0.000 | 0.000 | 0.000 | 0.000 | 0.000 | 0.000 | 0.000 | 0.000 |
| | N | 212 | 605 | 782 | 336 | 749 | 213 | 594 | 754 | 324 | 725 |
| IL, Israel | Pearson Corr. | 0.256 ** | 0.067 | 0.061 | −0.083 | | 0.431 ** | 0.417 ** | 0.353 ** | 0.232 ** | |
| | Sig. (2 tailed) | 0.000 | 0.318 | 0.286 | 0.177 | | 0.000 | 0.000 | 0.000 | 0.000 | |
| | N | 1176 | 226 | 310 | 266 | | 1158 | 230 | 293 | 265 | |
| IS, Iceland | Pearson Corr. | −0.055 | 0.039 | 0.193 ** | 0.074 | | 0.252 ** | 0.039 | 0.318 ** | 0.315 ** | |
| | Sig. (2 tailed) | 0.556 | 0.576 | 0.000 | 0.529 | | 0.006 | 0.576 | 0.000 | 0.006 | |
| | N | 117 | 210 | 430 | 74 | | 116 | 210 | 424 | 74 | |
| IT, Italy | Pearson Corr. | 0.279 ** | 0.439 ** | 0.172 ** | 0.276 ** | 0.297 ** | 0.494 ** | 0.558 ** | 0.414 ** | 0.377 ** | 0.621 ** |
| | Sig. (2 tailed) | 0.000 | 0.000 | 0.000 | 0.000 | 0.004 | 0.000 | 0.000 | 0.000 | 0.000 | 0.000 |
| | N | 264 | 139 | 827 | 1060 | 93 | 255 | 141 | 828 | 1053 | 93 |
| LT, Lithuania | Pearson Corr. | 0.337 ** | | 0.486 ** | 0.555 ** | | 0.394 ** | | 0.595 ** | 0.573 ** | |
| | Sig. (2 tailed) | 0.000 | | 0.000 | 0.000 | | 0.000 | | 0.000 | 0.000 | |
| | N | 675 | | 601 | 475 | | 665 | | 581 | 468 | |

**Table A2.** *Cont.*

| | | Correlations between Beliefs in Personal Ability to Limit Own Energy Use and Climate Change Mitigation Outcomes Expectation Variables | | | | | Correlations between Beliefs in Collective Ability to Limit Energy Use and Climate Change Mitigation Outcomes Expectation Variables | | | | |
|---|---|---|---|---|---|---|---|---|---|---|---|
| | | C | S | T | V | F | C | S | T | V | F |
| NL, Netherlands | Pearson Corr. | 0.084 | 0.123 | 0.040 | 0.114 ** | 0.034 | 0.272 ** | 0.025 | 0.202 ** | 0.239 ** | 0.302 ** |
| | Sig. (2 tailed) | 0.135 | 0.148 | 0.408 | 0.003 | 0.761 | 0.000 | 0.768 | 0.000 | 0.000 | 0.006 |
| | N | 316 | 139 | 431 | 662 | 83 | 311 | 140 | 436 | 663 | 82 |
| NO, Norway | Pearson Corr. | −0.002 | −0.059 | 0.059 | 0.070 | 0.017 | 0.112 | 0.257 ** | 0.168 ** | 0.170 ** | 0.172 ** |
| | Sig. (2 tailed) | 0.977 | 0.363 | 0.212 | 0.218 | 0.769 | 0.088 | 0.000 | 0.000 | 0.003 | 0.004 |
| | N | 234 | 241 | 445 | 308 | 284 | 232 | 240 | 445 | 308 | 286 |
| PL, Poland | Pearson Corr. | 0.151** | −0.111 | 0.202 ** | 0.234 ** | | 0.403 ** | 0.518 ** | 0.371 ** | 0.438 ** | |
| | Sig. (2 tailed) | 0.006 | 0.490 | 0.000 | 0.000 | | 0.000 | 0.001 | 0.000 | 0.000 | |
| | N | 324 | 41 | 461 | 645 | | 330 | 39 | 458 | 641 | |
| PT, Portugal | Pearson Corr. | 0.107 | 0.095 | 0.182 ** | 0.333 ** | | 0.218 ** | 0.304 ** | 0.222 ** | 0.255 ** | |
| | Sig. (2 tailed) | 0.125 | 0.234 | 0.000 | 0.000 | | 0.002 | 0.000 | 0.000 | 0.000 | |
| | N | 206 | 158 | 376 | 434 | | 199 | 157 | 365 | 415 | |
| RU, Russian Federation | Pearson Corr. | 0.500 ** | 0.429 ** | 0.466 ** | 0.375 ** | | 0.489 ** | 0.280 * | 0.662 ** | 0.669 ** | |
| | Sig. (2 tailed) | 0.000 | 0.000 | 0.000 | 0.000 | | 0.000 | 0.017 | 0.000 | 0.000 | |
| | N | 689 | 68 | 585 | 430 | | 685 | 72 | 623 | 411 | |
| SE, Sweden | Pearson Corr. | 0.098 | 0.060 | 0.211 ** | 0.159 * | 0.115 | 0.071 | 0.203 ** | 0.259 ** | 0.274 ** | 0.344 ** |
| | Sig. (2 tailed) | 0.222 | 0.280 | 0.000 | 0.013 | 0.144 | 0.336 | 0.000 | 0.000 | 0.000 | 0.000 |
| | N | 158 | 326 | 582 | 243 | 164 | 185 | 331 | 585 | 241 | 161 |
| SI, Slovenia | Pearson Corr. | 0.127 | 0.085 | 0.169 ** | 0.295 ** | 0.240 | 0.098 | 0.265 ** | 0.408 ** | 0.320 ** | 0.144 |
| | Sig. (2 tailed) | 0.109 | 0.349 | 0.005 | 0.000 | 0.056 | 0.222 | 0.003 | 0.000 | 0.000 | 0.245 |
| | N | 160 | 122 | 277 | 628 | 64 | 158 | 126 | 275 | 621 | 66 |

*: Correlation is significant at the 0.05 level (2-tailed); **: Correlation is significant at the 0.01 level (2-tailed).

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
