# Peer review of "The Challenges of Mitigating Climate Change Hidden in End-User Beliefs and Expectations"

_sustainability, doi:10.3390/su13052616_

Round 1

Reviewer 1 Report

The present paper explores challenges for reducing climate change, focusing on the beliefs and expectations of end-users. Here are several comments:

Comment 1. Abstract: The authors should re-edit this sentence for more fluency. “The conceptual proposition of the research, based on the concept of efficacy beliefs (considered both from personal and collective perspectives), states that decisions of the end-users to limit their energy use for reducing climate change could be motivated and thus explained by beliefs in personal and collective ability to contribute, i.e. to perform the actions required to achieve expected outcomes, and expectations that these personal and collective actions will result in anticipated benefits and outcomes for collective goal - i.e. climate change mitigation.”

Comment 2. Abstract: Instead of using general statements like “This opportunity could be addressed by relevant policy measures, providing more evidence of positive outcomes even from personal contributions and developing suitable means (e.g. smart platforms or some widely accepted social institutions) for collective contributions to increase awareness and belief in collective engagement”, mention and focus on a specific policy measure.

Comment 3. Abstract. “The authors should underscore the scientific value-added and the significance of their study in the Abstract.”

Comment 4. Introduction. The authors should edit this sentence to have better readability: “The conceptual proposition of the research, based on the concept of efficacy beliefs (considered both from personal and collective perspectives), states … i.e. climate change mitigation.”

Comment 5. Introduction. You cannot use the expression “on the other hand” [“On the other hand, belief in collective … is high.”] without having first “on the one hand”. 

Comment 6. Introduction. You have to include the objective(s) of the study, the aim is not enough [“The research aims to explore possible challenges explained by beliefs and expectations behind decisions of end-users to limit their energy use for climate change mitigation.”]

Comment 7. Literature review. This statement needs 1-2 references – “There is a widespread agreement that individual behavior would feasibly play a critical role in encouraging

 societal actions that mitigate climate change”

Comment 8. Literature review.  You probably wanted to write i.e., instead of “e.i” – “e.i. beliefs in personal and collective actions”.

Comment 9. Literature review. There are too many quotations in this chapter, you should try to reduce their number.

Comment 10. Materials and Methods. Are the questions from 1 to 3 the RQs of the study?

Comment 11. Materials and Methods Question 1 contains 4 questions, pls split them.

Comment 12. Materials and Methods. It is not clear if the matrix from Fig 3 (Uncertain, Inspired….) was created by you or was taken from literature?

Comment 13. Materials and Methods. Why have you established the limits for the two axes (Fig.4) to 2 and 8 in the Likert scale goes from 1 to 10?

Comment 14. Discussion. Remove “…” after policies  - “The challenges revealed, policies…”

Comment 15. Discussion. A paragraph with the limitation of the study is needed (several limitations presented in the Methodology chapter can be moved here)

Comment 16. Conclusion. Pls. introduce the Conclusion chapter with a paragraph.

Reviewer 2 Report

Dear Authors,

The present research, "the challenges for mitigating climate change hidden in end- user beliefs and expectations" shows the conceptual proposition of the research, based on the concept of efficacy beliefs. It is a very interesting study because it seeks to quantify the decrease in energy not by the use of active equipment but with social actions.

The methodology and results are adequate and interesting and can be used for action policies on energy use by users.

Author Response

# Reviewer2

The present research, "the challenges for mitigating climate change hidden in end- user beliefs and expectations" shows the conceptual proposition of the research, based on the concept of efficacy beliefs. It is a very interesting study because it seeks to quantify the decrease in energy not by the use of active equipment but with social actions.
The methodology and results are adequate and interesting and can be used for action policies on energy use by users.

+

Reviewer 3 Report

In the keywords a typo error here “ed user”

The first sentence in introduction requires stating page number as well next to reference as you are word by word adopting the question. Same applies for first sentence statement in page 2 Lorenzoni et al 2007. Kindly follow proper citation throughout entire manuscript.

The research question of this manuscript confuses the reader. The authors start with a question in beginning of introduction. In the methodology the authors present again questions that the study will cover, which are mostly of “what” format. These questions can be answered by the survey however the survey cannot  answer the “why” question. Further clarity in sequence of manuscript parts. Research question to be in end of introduction and need to be aligned with the method.

In page 2, the paragraph “the study outcomes,…” is somehow confusing as it stating findings here that appear to be confusing for the reader. Instead a problem statement can be described. Authors can build on that to support their approach in this paper. This paragraph need to be revisited.

The Literature review section is rich in theoretical concepts and background on various concepts that serve as core of the study. However, this is not revisited in the discussion section. The authors have to provide further linkage here to previous theories, findings in other studies.

Conclusion starts with numbered statements/findings. However, authors did not revisit the research question and assumed hypothesis. First paragraph should address these points and at later part authors can provide these “highlighted” findings. Limitation of the survey can be listed here, and insights for future research.

Reviewer 4 Report

  • Writing should be improved. Sentences are long, complex, and non-comprehensive.
  • Make appropriate use of a comma (,), colon, semicolon for long and complex sentences. “And” is used frequently and unnecessarily instead of “comma” in some places. Fix that.
  • Keywords: End-users.
  • Papers have been cited as [3} as well as (Lorenzoni et al., 2007). Is this a format of journal requirement? If not please be consistent.
  • Remove additional and unnecessary information inside the parenthesis. For example, in the abstract, the pre-industrial era is defined inside the parenthesis, which is irrelevant for this paper. If someone wants to understand more about the pre-industrial era, they can read the original document. Same thing at beginning of the second paragraph of the introduction section. This is a trend along with the paper, thus making the paper unclear and non-comprehensive. Please reduce noises.
  • “This research attempts to clarify the ties…”: Wasn’t the theory (or ties) adopted by this paper was clear before? Is it clarifying previous the theory or addressing confusion about previous research?
  • Sentences are long. Break them and make them more comprehensive. Also, information is repetitive. For example, the phrase “efficacy beliefs (considered both from personal and collective perspectives)” is repeated in several locations. Also, You can define efficacy belief in the context of your paper and use that term on the rest of the paper without making it too long.
  • Why only end-users are emphasized in this paper? What is the significance of emphasizing “end-users” such that it becomes a keyword for the paper? Why not other users?
  • The second last paragraph of the introduction section talks about the finding of research which is unnecessary and irrelevant in the introduction section. The introduction should introduce the problem and need of your research, but not the result.
  • The first four paragraphs of the literature review and theoretical background are mostly irrelevant for introducing the model which you want to adopt for the paper. Reduce words and present what is only needed to justify your research. The theoretical consideration is mostly explained on page 5 and six.
  • REDUCE IRRELEVANT INFORMATION FROM THE PAPER. REDUCE LENGTH. 
  • NO MAJOR EDITS.

Reviewer 5 Report

The paper explores potential challenges for reducing climate change. In particular, authors conduct a detailed empirical descriptive analysis on the impact of current urbanization patterns on dominating type of social environment and level of inspiration/motivation built, based on beliefs and expectations in personal and collective ability to contribute to climate change mitigation and positive outcomes expectations from personal and collective contribution. This analysis is carried out in an orderly way and the reader can follow the rationale of the essay, also because of a plain and easy English style and grammar. Therefore, the research results in a very interesting work for researchers, practitioners and policy-makers.

However, this reviewer thinks it can be further improved by considering the following suggestions.

Please be aware that reference 3 is not only "Lorenzoni", but it is "Lorenzoni et al."

P.4: You wrote "Finlay the collective efficacy was operationalised as belief in collective ability to engage and contribute to climate change mitigation by limiting energy use", do you mean "Finally...."?

Conclusion is currently scant. It clearly presents the main findings, but it is also supposed to draw together the main points given in the paper to explain how they connect and relate and to place the concepts in a somewhat wider context. It should also provide suggestions for further research. The bullet point structure should at least be introduced by an explanatory sentence that anticipates what the points will be about. It would be better to restructure the section so that it flows more smoothly.

Round 2

Reviewer 1 Report

Lines 255-257: "To measure the average levels of indicators defining the degrees of beliefs in personal and collective abilities to contribute to climate-change mitigation and positive outcome expectations in the surveyed countries and various domicile categories": It will be better to rephrase this objective, it doesn't read well.

Author Response

Dear Reviewer,

Thank you for your remarks. The sentence in lines 255-257 has been corrected and written as follows:

"To measure beliefs in personal and collective abilities to contribute to climate-change mitigation and positive outcome expectations in the surveyed countries and various domicile categories"

Best regards

Gerda Zigiene